# Electrostatic and steric effects underlie acetylation-induced changes in ubiquitin structure and function

Simon Maria Kienle[1,2], Tobias Schneider[2,3], Katrin Stuber[1,2,3], Christoph Globisch[3], Jasmin Jansen[1,2], Florian Stengel [1,2], Christine Peter [2,3], Andreas Marx [2,3] ✉, Michael Kovermann [2,3] ✉ & Martin Scheffner [1,2] ✉

Covalent attachment of ubiquitin (Ub) to proteins is a highly versatile posttranslational modification. Moreover, Ub is not only a modifier but itself is modified by phosphorylation and lysine acetylation. However, the functional consequences of Ub acetylation are poorly understood. By generation and comprehensive characterization of all seven possible mono-acetylated Ub variants, we show that each acetylation site has a particular impact on Ub structure. This is reflected in selective usage of the acetylated variants by different E3 ligases and overlapping but distinct interactomes, linking different acetylated variants to different cellular pathways. Notably, not only electrostatic but also steric effects contribute to acetylation-induced changes in Ub structure and, thus, function. Finally, we provide evidence that p300 acts as a position-specific Ub acetyltransferase and HDAC6 as a general Ub deacetylase. Our findings provide intimate insights into the structural and functional consequences of Ub acetylation and highlight the general importance of Ub acetylation.

The covalent modification of proteins by ubiquitin (Ub) involves three classes of enzymes, Ub-activating enzymes E1, Ub-conjugating enzymes E2, and Ub-protein ligases E3, with E3 ligases mediating the substrate specificity of the Ub-conjugation system[1,2]. In most cases, Ub is conjugated to lysine residues of substrate proteins via isopeptide bond formation, a process that is reversible by the action of deubiquitylating enzymes (DUBs)[3]. Since Ub can serve as its own substrate and since any of its 7 lysine residues and the α-amino group of the N-terminal methionine can be ubiquitylated, a virtually indefinite number of different types of Ub chains can be formed, making ubiquitylation one of the most versatile post-translational modifications in eukaryotes[4]. Notably, different types of ubiquitylation have different effects on their target proteins. For instance, attachment of a single Ub moiety (mono-ubiquitylation) can affect the activity or subcellular localization of a protein[5], while K48-linked Ub chains target substrates

for proteasomal degradation[1] and K63-linked and linear, M1-linked Ub chains have been associated with signal transduction pathways and selective autophagy[6,7].

In many cases, the different types of ubiquitylation, also referred to as Ub code[4], are recognized by distinct Ub-binding proteins, which thereby play a decisive role in defining the eventual fate of ubiquitylated proteins[8]. In a simplified view, Ub-binding proteins contain distinct Ub-binding domains (UBDs) or motifs, enabling the interaction with ubiquitylated proteins, and via a second protein interaction site, mediate the interaction of ubiquitylated proteins with downstream effectors[8,9]. For instance, proteins such as HHR23A/B and Ubiquilin-1–4 harbor a Ub-associated (UBA) domain and a Ub-like (Ubl) domain, thereby shuttling ubiquitylated proteins to the proteasome[9–11]. Another example is NDP52 (also known as Calcoco 2), which participates in xenophagy of intracellular pathogens[7,12]. NDP52 contains a

[1]Department of Biology, University of Konstanz, Konstanz, Germany. [2]Konstanz Research School Chemical Biology, University of Konstanz, Konstanz, Germany. [3]Department of Chemistry, University of Konstanz, Konstanz, Germany. ✉e-mail: andreas.marx@uni-konstanz.de; michael.kovermann@uni-konstanz.de; martin.scheffner@uni-konstanz.de

C-terminal Ub-binding zinc finger (ZF) and a non-canonical LIR motif for LC3C binding, thereby acting as an adaptor between ubiquitylated cargos and phagophores with lipidated Atg8 family proteins[12].

Recently, NDP52 was also described to play a role in PARKIN-mediated autophagic clearance of mitochondria (mitophagy)[13,14]. In mitophagy, phosphorylation of S65 of Ub plays a critical role, as phosphorylated Ub stimulates both the E3 ligase activity of PARKIN and PINK1-mediated phosphorylation and activation of PARKIN[15,16]. This showed that Ub is not only a modifier but is subject to post-translational modification itself, adding yet another layer of complexity to the Ub code[17]. In addition to phosphorylation, mass spectrometric approaches have revealed that with the potential exception of K29, any of the lysine residues of Ub can be acetylated[17]. Furthermore, the level of Ub acetylation is affected by inhibition of deacetylases and, more importantly, by stress stimuli, including ionizing radiation and induction of autophagy[17–19]. However, besides the observations that acetylation of Ub at K6 or K48 interferes with Ub chain elongation by certain E2s[19] and that acetylation of Ub can somehow stabilize the mono-ubiquitylated form of histone H2B[19], nothing is known about the structural and functional consequences of Ub acetylation.

Here, by employing genetic code expansion[20], we generated seven Ub variants that are site-specifically acetylated at one of the seven lysine residues. NMR spectroscopic analysis reveals for each of the mono-acetylated variants unique structural changes, the extent of which depends on the site of acetylation. Furthermore, the structural differences are reflected in different usage of the Ub variants by the E3 ligases tested. Therefore, we used the acetylated Ub variants as bait molecules in an affinity enrichment coupled to high-resolution mass spectrometry (AE-MS) approach and identified proteins that interact with Ub in acetylation and site-specific manner, such as NDP52 that selectively binds to Ub with an acetyllysine at position 6. Notably, we provide strong evidence that the replacement of lysine by the

frequently used acetyllysine surrogate glutamine does not faithfully mirror the structural and functional properties of some of the truly acetylated Ub variants. Finally, we show that, as suggested by published in vivo data[21], the acetyltransferase p300 modifies Ub at distinct lysine residues in vitro, while HDAC6 deacetylates acetylated Ub variants in a position-independent manner.

## Results

### Generation of site-specifically acetylated ubiquitin variants and their performance in in vitro ubiquitylation assays

A prerequisite to study the effects of acetylation on Ub structure and function is the generation of homogeneous populations of site-specifically acetylated Ub. To do so, we employed the method of genetic code expansion, since it allows us to incorporate acetyllysine (AcK) at any desired position within a protein of interest in bacteria[20]. Accordingly, we generated seven site-specifically acetylated Ub variants by replacing one of the seven lysine residues of Ub with AcK (Ub xAcK, where x stands for the respective lysine position within Ub) (Fig. 1a). For completeness, we also included Ub 29AcK, although acetylation of K29 has not yet been reported[17]. Quantitative incorporation of AcK into Ub and the homogeneity of the seven purified Ub AcK variants were confirmed by ESI-MS and MS/MS analysis (Supplementary Fig. 1).

With the different Ub AcK variants in hand, we tested whether they can substitute for nonmodified Ub in autoubiquitylation assays using two RING E3 Ub ligases (HDM2 and RLIM) and a HECT E3 (E6AP/UBE3A)[22,23]. In agreement with previous reports, Ub 6AcK and Ub 48AcK were less efficiently used than nonmodified Ub, irrespective of the E3 employed (Fig. 1b,c; Supplementary Fig. 2)[19]. Furthermore, while Ub 29AcK, Ub 33AcK, and Ub 63AcK did not appear to have a significant effect on the autoubiquitylation capacity of the E3s tested, Ub 11AcK and Ub 27AcK hampered autoubiquitylation, though to

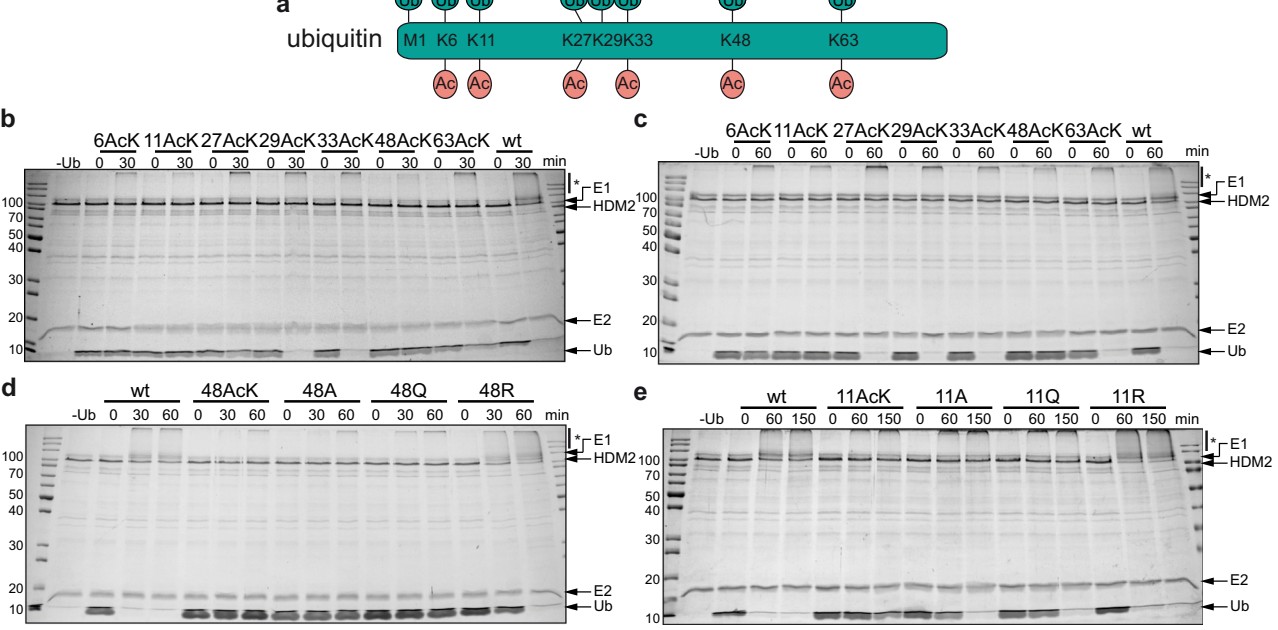

**Fig. 1 | Acetylated Ub variants and their performance in HDM2 autoubiquitylation. a** Lysine residues of ubiquitin (Ub) used for Ub chain formation and acetylation (Ac) in cells. **b, c** Ub variants acetylated at K6 (6AcK), K11 (11AcK), K27 (27AcK), or K48 (48AcK) are inefficiently used by HDM2 for autoubiquitylation. **d** Autoubiquitylation assays with Ub variants, in which K48 was replaced by A, Q, or R reveal that the presence of a positive charge at position 48 is critical for efficient HDM2 autoubiquitylation. **e** Autoubiquitylation assays with Ub variants, in which K11 was replaced by A, Q, or R, indicate that the inefficient use of Ub 11AcK by HDM2 for autoubiquitylation cannot be explained by charge neutralization only.

**b–e** Autoubiquitylation reactions were performed as described in Methods and started by addition of the respective Ub variants or nonmodified Ub (wt). Reactions were stopped at the times indicated. Reactions in the absence of Ub (−Ub) were stopped at 30 min (**b**), 60 min (**c, d**), or 150 min (**e**). All reactions were analyzed by SDS-PAGE followed by Coomassie blue staining. Running positions of molecular mass markers (kDa), HDM2, UBA1 (E1), UbcH5b (E2), free Ub (Ub), and autoubiquitylated forms of HDM2 (*) are indicated. The experiments shown are representative of three independent experiments. Source data are provided as a Source Data file.

somewhat different extents (Ub 48AcK > Ub 11AcK > Ub 27AcK for HDM2 and RLIM; Ub 48AcK > Ub 27AcK > Ub 11AcK for E6AP) (Fig. 1b, c; Supplementary Fig. 2). In line with previous reports, the different efficiencies of the Ub AcK variants in autoubiquitylation are due to intrinsic properties of the E3s used rather than to differences in Ub activation and/or the ability of the respective Ub-conjugating enzymes to accept activated Ub (Supplementary Fig. 2)[19].

The inability of E6AP to utilize Ub 48AcK for autoubiquitylation is readily explained by the notion that E6AP mainly catalyzes the formation of K48-linked Ub chains[24]. However, this argument cannot be applied to the other AcK variants and may also not be applicable to HDM2, since it can use any lysine residue in Ub chain formation[23,25]. A general effect of lysine acetylation is the elimination of the positive charge of the ε-amino group of the lysine side chain[26]. To investigate whether this mechanism accounts for the observed effects, we replaced respective lysine residues with arginine, glutamine (which is commonly used in mutational studies as an AcK surrogate[19,27–30]), and alanine, respectively, and performed autoubiquitylation assays. For HDM2, we found that a positive charge at position 48 is sufficient for proper autoubiquitylation, since, unlike Ub 48A, Ub 48Q, or Ub 48AcK, Ub 48R was efficiently used for autoubiquitylation (Fig. 1d). In contrast, E6AP did not use any of the Ub variants as expected (Supplementary Fig. 2). Similar results were obtained with Ub variants modified at position 6, 11, or 27 for both E3s, insofar as a positive charge at these positions appears to be an important determinant for efficient autoubiquitylation (Fig. 1e; Supplementary Fig. 2).

The above results clearly indicate that acetylation of K6, K11, K27, and K48 interferes with efficient Ub chain formation, which is, at least in part, caused by charge neutralization. For K11, this effect is explained on a structural level by the fact that the side chain amino group is involved in a salt bridge with E34. However, features other than charge neutralization appear to be involved as well. For instance, Ub 11A and Ub 11Q are more efficiently used by HDM2 for autoubiquitylation than Ub 11AcK (free Ub 11A and Ub 11Q are completely consumed after a reaction time of 150 min, while free Ub 11AcK can still be observed; Fig. 1e). Moreover, the results obtained with the Ub 11 variants support previous reports[31,32] indicating that the frequently used AcK surrogate Q does not reliably mimic all inherent features of AcK.

### Acetylation affects Ub structure in a site-specific manner

Because of the different behavior of the Ub AcK variants in autoubiquitylation assays, we wondered if these differences are mirrored on the structural level. Therefore, we generated isotopically labeled Ub AcK variants to conduct high-resolution NMR spectroscopy. For all seven Ub AcK variants, two-dimensional $^{1}$H–$^{15}$N heteronuclear single quantum coherence (HSQC) NMR spectra were acquired (Supplementary Fig. 3), and weighted chemical shift perturbations (CSPs) were calculated for backbone amide resonances referenced to nonmodified Ub (Fig. 2). In this way, changes in the chemical environment can be unraveled which are caused by structural rearrangements of Ub as a consequence of acetylation of a specific lysine residue.

Unlike phosphorylation of Ub at S65[16,33], acetylation of any of the lysine residues did not result in the appearance of additional cross peaks in the $^{1}$H–$^{15}$N HSQC spectra, indicating that population equilibria are not affected on the slow NMR time scale (Supplementary Fig. 3). However, CSP analysis revealed that acetylation of respective lysine residues induces distinct CSP patterns. Strong perturbations in terms of amplitude and sequence distribution are observed for Ub 11AcK, Ub 27AcK, Ub 29AcK, and Ub 33AcK (Fig. 2b–e). In contrast, clusters of significant CSP values for Ub 6AcK, Ub 48AcK, and Ub 63AcK are exclusively located in spatial proximity to the actual modification site and are relatively weak in the case of the latter two (Fig. 2a, f, g). Note that no spectral information is available for the acetylated lysine residues, due to the non-isotopically labeled backbone amide nitrogen.

Ubiquitylation/Ub chain elongation is facilitated by binding of E2 conjugating enzymes and/or E3 ligases to distinct surface areas of Ub, which usually comprise residues of the I44 (L8, I44, H68, and V70) and I36 (L8, I36, L71, and L73) hydrophobic surface patches[34–37]. Interestingly, only acetylation of K11 and K27 significantly affects the residues of these hydrophobic patches (Fig. 2b, c), indicating that the side chains of K11 and K27 are of particular importance for the conformational plasticity of Ub. In contrast, for Ub 29AcK and Ub 33AcK significant CSP values are observed at the central α-helix and the β1- or β2-strand, which are in direct contact with the corresponding lysine side chain, but do not encompass hydrophobic patch residues (Fig. 2d, e).

The observations from the CSP analysis are in good agreement with the results obtained in the autoubiquitylation experiments. For instance, we only detected local structural alterations for Ub 6AcK and Ub 48AcK, and their repressive effect on E3 ligase function was due to the loss of the positive charge at the respective position (Fig. 1d; Supplementary Fig. 2). For Ub 11AcK, not only the charge but also the composition of the side chain has an impact on E3 activity (i.e., Q cannot fully mimic the effect of AcK); in this case, acetylation is accompanied by disruption of the I44 and I36 hydrophobic patches that are essential for efficient interplay between the E2/E3 enzyme complex and Ub[34–37].

### Ub acetylated at K11 is structurally different from its glutamine surrogate

If the effects of K11 acetylation on Ub structure and on E3 activity are indeed interconnected, we can postulate that replacement of K11 by the AcK mimic Q has different effects on Ub conformation than acetylation of K11. To address this hypothesis, we acquired $^{1}$H–$^{15}$N HSQC NMR spectra of $^{15}$N isotopically labeled Ub variants 11Q, 11A, and 11R (Fig. 3a) and calculated CSPs with reference to nonmodified Ub (Fig. 3b–e). Ub 48R and Ub 48Q served as further control (Supplementary Fig. 4).

For Ub 11AcK, we identified three clusters of residues with significant perturbations (Fig. 3b). These are located (i) in the β2-strand, (ii) at the C-terminal end of the central α-helix, and (iii) at the transition from the β5-strand to the tail region. Cluster (i) can be ascribed to next neighbor effects at the modification site, and cluster (iii) is a propagation of the structural effect on the hydrophobic patches as described above. The perturbations regarding cluster (ii) in the α-helix are likely caused by disruption of the salt bridge between the side chains of K11 and E34, which is an inevitable consequence when the positive charge is missing[38,39]. However, the latter is not only the case for lysine acetylation but also for lysine replacement by glutamine or alanine. Yet, in the CSP mappings of Ub 11Q and Ub 11A, the α-helical part is hardly affected (Fig. 3c, d). Therefore, we concluded that an additional feature of AcK is required to achieve the entire structural effect observed for Ub 11AcK. Comparing AcK with Q and A, a major difference is the length of the side chain. Attachment of an acetyl group to the K side chain (7.8 Å in length) results in a total length of approximately 10.6 Å. This is almost twice as long as the Q side chain (6.1 Å) and seven times longer than the methyl group of alanine (1.5 Å). Thus, we propose that in addition to electrostatic effects (i.e., neutralization of the positive charge), the side chain at position 11 has to be long enough to have a steric effect on the residues in the opposing α-helix. This hypothesis is supported by several lines of evidence. Firstly, for Ub 11R, which showed similar efficiencies in the autoubiquitylation assay as nonmodified Ub (Fig. 1e; Supplementary Fig. 2), no significant impact on the α-helix could be detected (Fig. 3e). Thus, the R side chain, which is 9.0 Å in length, does not appear to be long enough to affect the α-helix and—as evidenced by the NMR solution structure of lysine-free (K0) Ub (PDB ID 2MI8)[40]—is still able to form the salt bridge due to its positive charge. Secondly, in molecular dynamics (MD) simulations of the respective Ub K11 variants, we additionally observed that 11A and 11Q are further apart from the opposing E34 side chain than 11AcK

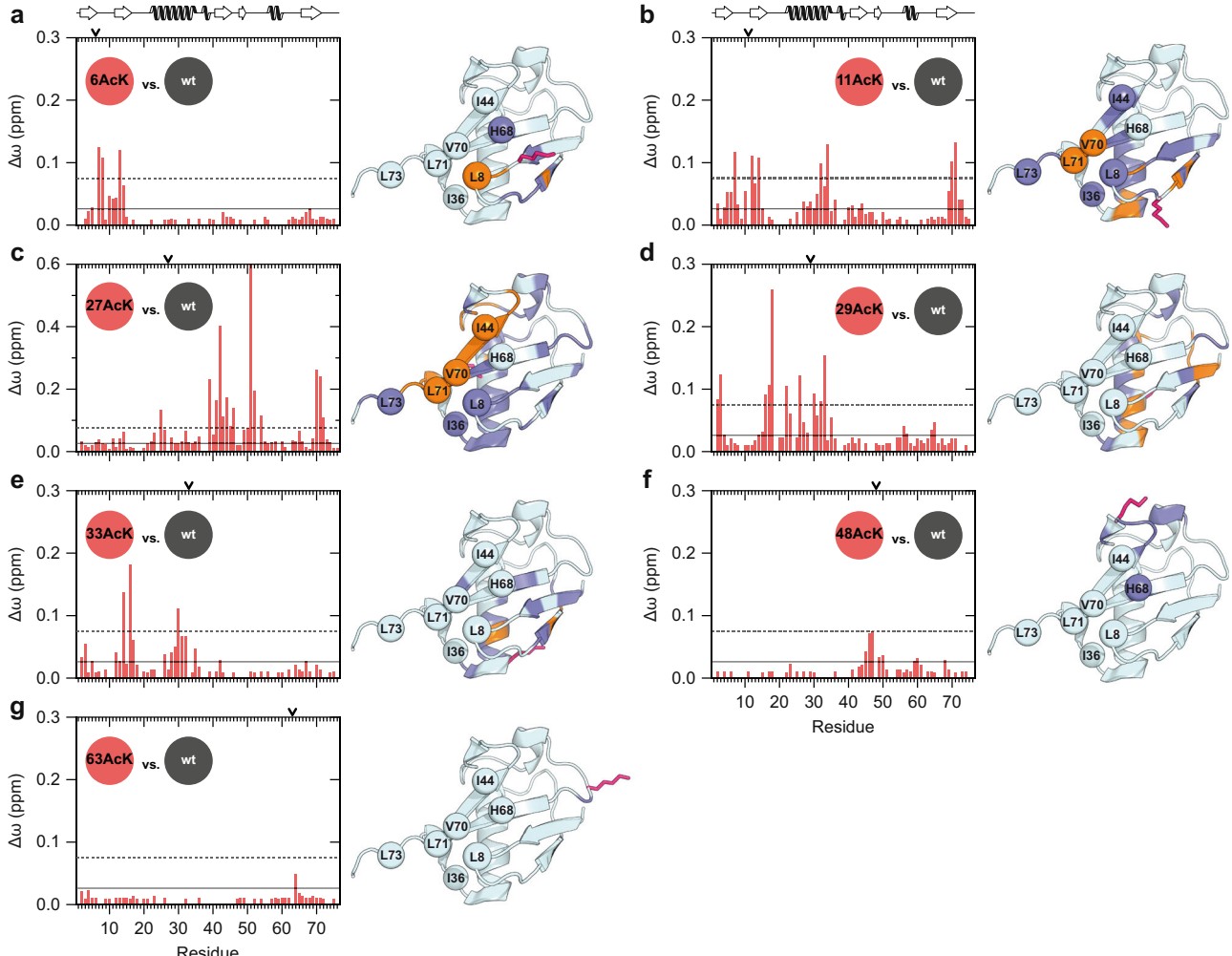

**Fig. 2 | Site-specific Ub acetylation results in distinct structural changes.**
**a–g** Weighted chemical shift perturbation (CSP, Δω) mappings of backbone amide resonances are shown comparing the different mono-acetylated Ub variants each with nonmodified Ub (wt). CSP plots are depicted on the left for Ub 6AcK (**a**), Ub 11AcK (**b**), Ub 27AcK (**c**), Ub 29AcK (**d**), Ub 33AcK (**e**), Ub 48AcK (**f**), and Ub 63AcK (**g**), whereas significant Δω values are highlighted on the NMR solution structure of nonmodified Ub (wt) (PDB ID 1D3Z)[79] presented in cartoon mode on the right. Note the different scaling used in the CSP mapping of Ub 27AcK in **c** in contrast to the other graphs. The cutoff values are the same in all plots and were calculated by taking the mean (horizontal solid line) and the mean plus one standard deviation

(horizontal dashed line), respectively, over all Δω values from **a** to **g** excluding values from lysine residues. Residues with Δω values above the solid line are colored in slate blue and above the dashed line are colored in orange in the structure. CA atoms of residues being part of the I44 (L8, I44, H68, and V70) or the I36 hydrophobic patches (L8, I36, L71, and L73) are drawn as spheres in the structure and are labeled by use of the one letter code. Secondary structural elements of Ub are schematically illustrated on the top of the plots in **a** and **b**. The respective lysine residue which is modified by acetylation is indicated by an arrow on the top of all CSP plots and the side chain is drawn as stick model in pink in the structure. Source data are provided as a Source Data file.

(Supplementary Fig. 4). Thirdly, a similar CSP pattern as with Ub 11AcK is observed when the side chain of K11 is elongated by addition of a Boc group[41].

In contrast to the K11 variants, the CSP patterns of Ub 48AcK, Ub 48Q, and Ub 48R are virtually identical, showing only one cluster of significant values around the modification site (Supplementary Fig. 4). As the interference with HDM2 autoubiquitylation is comparable between Ub 48AcK and Ub 48Q, while Ub 48R has a rather mild effect (Fig. 1d), mainly the charge but not the length of the side chain appears to be of critical importance in this case.

## Acetylation of Ub at K11 affects its binding properties

To further prove our hypothesis that not only the loss of the basic charge but also the length of the side chain is important for the effect of K11 acetylation on Ub properties, we performed interaction studies monitored by NMR spectroscopy with the UBA domain of human Ubiquilin-2 (also referred to as Chap1 or hPLIC-2)[10,11]. Like Ubiquilin-1, Ubiquilin-2 binds to an area of Ub that comprises the I44 hydrophobic

patch[11]. Moreover, unlike most other UBA domains, the UBA domains of Ubiquilin-1/−2 (their UBA domains differ by only one amino acid) bind to free Ub with a dissociation constant, $K_d$, in the low micromolar range[10,11] and, thus, are well suited to study the interaction with Ub variants.

The unlabeled UBA domain of Ubiquilin-2 was titrated stepwise to $^{15}N$ isotopically labeled Ub 11AcK, Ub 48AcK, Ub 11Q, and Ub 48Q as well as nonmodified Ub as a reference to determine individual binding interfaces and affinities (Fig. 4; Supplementary Fig. 5). This showed that all Ub variants studied display similar CSP patterns upon UBA binding (Fig. 4a; Supplementary Fig. 5), which are in line with the reported CSP patterns of nonmodified Ub interacting with the UBA domain[11,42]. Thus, we conclude that the UBA domain binds to all Ub variants studied in a similar fashion via the I44 patch. However, the binding affinity of the UBA domain for Ub 11AcK ($K_d = 8.0 \pm 0.8\,\mu M$) is significantly weaker than for nonmodified Ub ($K_d = 2.0 \pm 0.4\,\mu M$), while the other Ub variants, including Ub 48AcK and Ub 11Q have dissociation constants comparable to that of nonmodified Ub (Supplementary Fig. 5). This

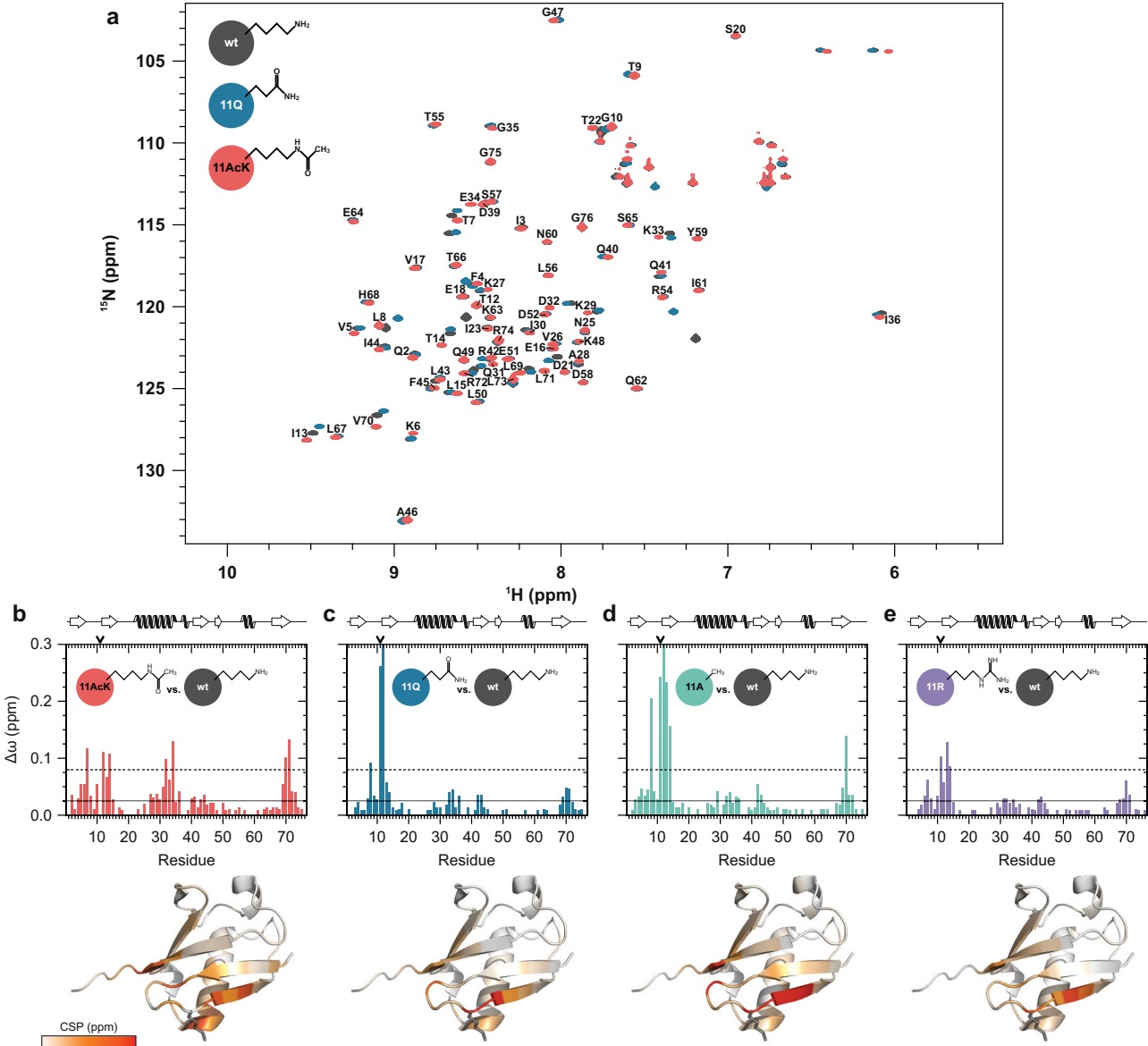

**Fig. 3 | Ub 11AcK and Ub 11Q are structurally different. a** Superimposition of the two-dimensional heteronuclear $^1H$–$^{15}N$ HSQC NMR spectra of Ub 11AcK (red), Ub 11Q (blue) and nonmodified Ub (wt, gray), each with the respective side chain indicated. The labeling of the backbone amide resonances by the one letter code corresponds to the spectrum of Ub 11AcK. **b**–**e** Weighted chemical shift perturbation (CSP, Δω) mappings are shown for Ub 11AcK (**b**), Ub 11Q (**c**), Ub 11A (**d**), and Ub 11R (**e**) versus nonmodified Ub (wt). The cutoff values are the same for all plots and were calculated by taking the mean (horizontal solid line) and the mean plus one standard deviation (horizontal dashed line), respectively, over all Δω values from

(**b**) to (**e**) excluding values from the respective residue at position 11. Secondary structural elements according to the NMR solution structure of nonmodified Ub (PDB ID 1D3Z)[79] are schematically depicted on top of all graphs. The same structure is used in the lower panel to illustrate the location and amplitude of the perturbations with colors from white to red in continuous mode. The side chain of K11 is drawn as stick model in gray in the structure and the position in the sequence is indicated by an arrow on top of the graphs. Source data are provided as a Source Data file.

finding confirms our hypothesis that, in contrast to its glutamine surrogate, acetylation of K11 alters the shape of the I44 hydrophobic patch thereby modulating the binding properties of Ub. In this context, it should be noted that the amide cross peak corresponding to K48 experiences the highest perturbation and extensive line broadening in the $^1H$–$^{15}N$ HSQC NMR spectrum during UBA titration, indicating that K48 is crucially involved in the interaction with the UBA domain (Fig. 4a, c; Supplementary Fig. 5). Yet, acetylation of its side chain does not significantly affect the binding affinity of Ub for the UBA domain, which is explained by the fact that K48 acetylation does not induce structural changes in the I44 hydrophobic patch region (Fig. 2f).

Taken together, neutralization of the positive charge of a lysine side chain by acetylation is crucial to induce structural rearrangements

in Ub, but depending on the actual position, the length of the side chain is also crucial to exert the full effect. Because of the latter, glutamine, which is frequently used in mutational studies to mimic AcK[19,27–30], is a poor surrogate for AcK in certain cases and, thus, in the absence of structural data, results obtained with respective "K-to-Q" protein variants should be interpreted with caution.

## Identification of the interactome of acetylated Ub variants

Since all seven Ub AcK variants adopt distinct conformations and since acetylation of K11 modulates the binding of Ub to a UBA domain, we wondered if acetylation in general influences the protein-protein interaction properties of Ub. Hence, AE-MS experiments were employed with whole cell extracts. To do so, we equipped all Ub AcK

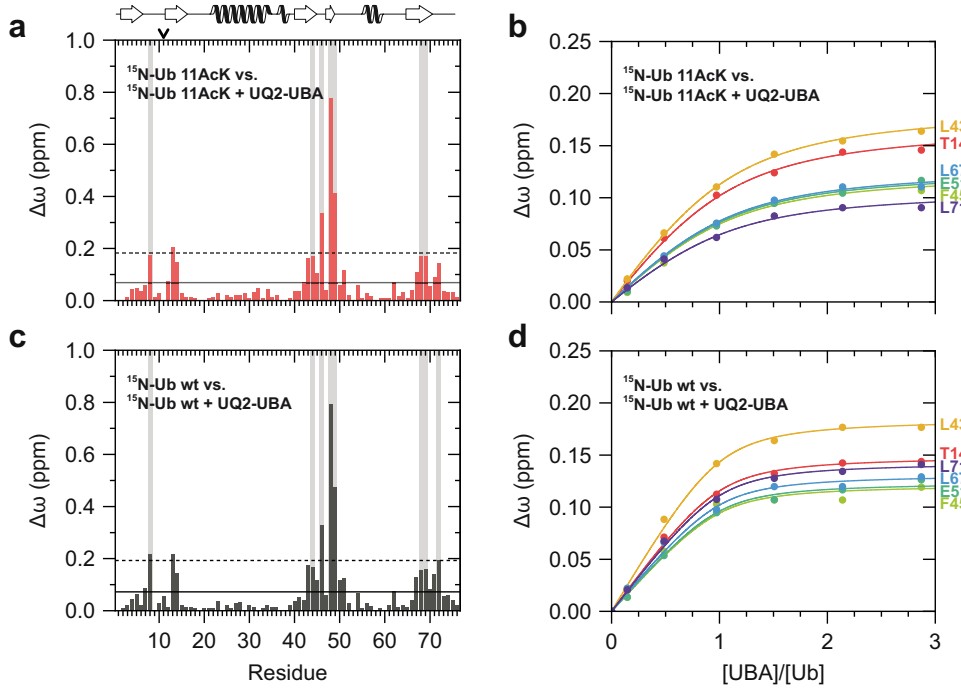

**Fig. 4 | Structural differences between Ub 11AcK and nonmodified Ub result in different binding affinities for the UBA domain of Ubiquilin-2. a, c** Weighted chemical shift perturbation (CSP, Δω) mapping obtained at the endpoint of the NMR titration experiment of Ub 11AcK (**a**) and nonmodified Ub (**c**) with the UBA domain of Ubiquilin-2 (UQ2-UBA) in approximately three times molar excess. Values exceeding the horizontal solid line are higher than the mean, and values exceeding the horizontal dashed line are higher than the mean plus one standard deviation. Background lines in gray highlight residues experiencing more than 90% decrease of signal intensity during the titration due to slow or intermediate exchange. Secondary structural elements are depicted on the top of the graph according to the NMR solution structure of nonmodified Ub (PDB ID 1D3Z)[79]; the position of 11AcK is indicated by an arrow in **a. b, d** Individual binding isotherms are shown for the residues that were included into the global fitting procedure in order to determine the dissociation constant of the complex of UQ2-UBA with Ub 11AcK (**b**) and nonmodified Ub (**d**), with a concentration of Ub 11AcK and Ub of $c = 33\,\mu M$. The corresponding $K_d$ values are provided in the text and in Supplementary Fig. 5. Source data are provided as a Source Data file.

variants with an N-terminal Strep-tag II for immobilization on Strep-Tactin® beads. Furthermore, nonmodified Ub was used for comparison and empty beads as control. In brief, the various bait molecules were incubated with whole cell extracts derived from HEK293T cells. Affinity-enriched proteins were identified by label-free quantitative MS and significantly enriched proteins were determined by ANOVA statistics (Fig. 5a).

Hierarchical clustering revealed 249 significant interactors with preferred binding to a single or several Ub AcK variants (Fig. 5b). We found proteins that preferentially interact with Ub 48AcK (Cluster 5), with Ub 11AcK (Cluster 4), and with Ub 48AcK, Ub 11AcK, and Ub 27AcK (Cluster 1) or that show no preference for any Ub variant (Cluster 7). Moreover, since acetylation of K11 had a unique structural impact on Ub, we also found proteins with decreased binding to this Ub variant, while they were similarly enriched for all other Ub variants (Cluster 3 and 6). Finally, proteins were identified that preferentially interact with Ub 29AcK, Ub 33AcK, Ub 63AcK, and nonmodified Ub (Cluster 2). In line with this, these 4 Ub variants show a medium-high correlation in their overall binding properties (Fig. 5c), which also correlates with their behavior in autoubiquitylation assays (Fig. 1b,c, Supplementary Fig. 2).

Gene ontology (GO) term enrichment analysis of the interactomes shows that proteins involved in membrane fusion were enriched with Ub 11AcK, Ub 48AcK, and Ub 27AcK (Fig. 5d, Cluster 1). Components of the BRCC36 (encoded by the *BRCC3* gene) isopeptidase-containing complex (BRISC), which hydrolyzes K63-linked Ub chains[43], were enriched with Ub 29AcK, Ub 33AcK, Ub 63AcK, and nonmodified Ub (Fig. 5d, Cluster 2), whereas subunits of the CTLH complex, which was recently shown to function as a multi-subunit Ub E3 ligase[44], were enriched for all Ub variants except for Ub 11AcK (Fig. 5d, Cluster 6).

Interestingly, besides BRISC, which belongs to the JAMM/MPN+ DUB family[3], we identified 17 DUBs (8 of these belong to the USP family) with a distinct binding preference for certain Ub variants (Supplementary Fig. 6). Moreover, components of the 19 S regulatory particle of the 26 S proteasome were enriched with Ub 48AcK (Fig. 5d, Cluster 5). Since the regulatory particle preferentially binds K48-linked Ub chains[6,45,46], we wondered if, in general, acetylation of Ub, which results in the formation of an amide bond, may in part resemble isopeptide bond-linked Ub dimers. Therefore, we structurally compared the seven Ub AcK variants with the proximal Ub of the respective dimer[47]. Interestingly, we found a high similarity in the CSP patterns for all Ub AcK variants and the proximal Ub of the respective dimer, except for Ub 6AcK and Ub 48AcK (Supplementary Fig. 6). The proximal unit of these Ub dimers showed additional signals, which result from intramolecular interactions between the proximal and distal moieties[47]. Nonetheless, our data indicate that acetylated Ub mimics at least in part the proximal unit of a Ub dimer, or in other words, lysine acetylation exerts virtually the same structural influence on Ub as isopeptide-conjugation of another Ub molecule does (see also Discussion).

### NDP52 binds to Ub acetylated at position 6 via its zinc finger

To corroborate the results obtained by the AE-MS approach, we validated some interactions via western blot analysis after affinity enrichment. Here we found a high correlation between the interaction patterns obtained by AE-MS and western blot analysis (Fig. 6a, b). Besides proteins that potentially mediate the consequences of Ub acetylation, the identification of proteins/enzymes involved in regulating Ub acetylation is of importance. Strikingly, with the exception of HDAC6 (Fig. 6a, b), our interactome analysis did not identify any

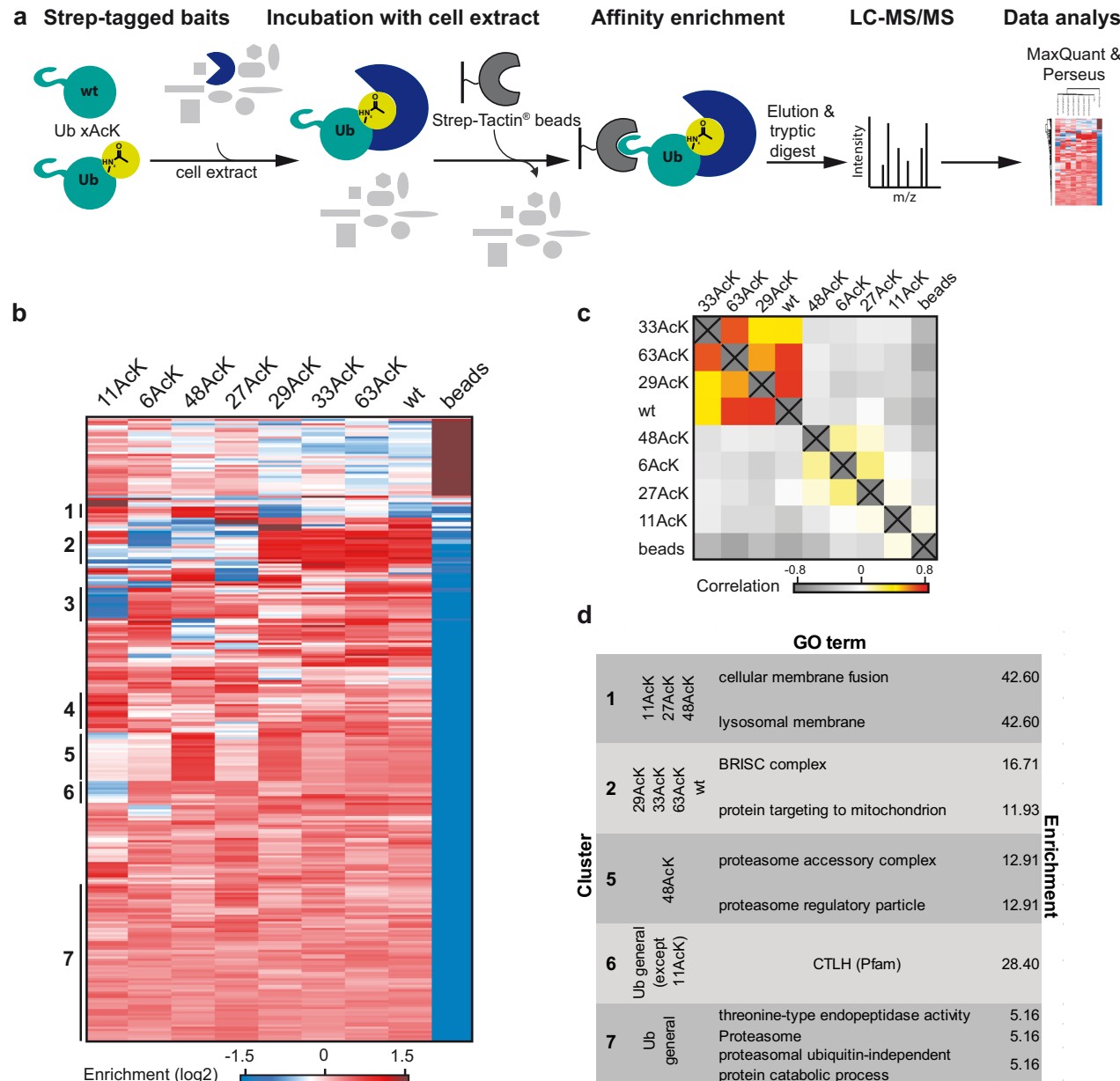

**Fig. 5 | Identification of interactors of acetylated Ub variants. a** Schematic overview of the AE-MS workflow for the identification of interactors of acetylated Ub variants. Nonmodified Ub (wt) and each of the seven site-specifically acetylated Ub variants (Ub xAcK) were used as bait molecules for affinity enrichment and incubated with HEK293T cell extract. Empty beads were used as control. **b** Hierarchical clustering of statistically significant interactors (rows) of the different bait molecules (columns). Red indicates enrichment, whereas blue indicates lack of enrichment. Thresholds for ANOVA statistics were set to FDR = 0.002 and $S_0 = 4$. Cluster numbers are indicated on the left. **c** Correlation heatmap of significantly enriched interactors between the different acetylated Ub variants is plotted. Strong correlation is shown in red, medium correlation in yellow, no correlation in white, and inverse correlation in gray. **d** GO terms, KEGG and Pfam annotations for the respective clusters identified in **b**. Source data are provided as a Source Data file.

known acetylating and deacetylating enzymes. With respect to the latter, it should be noted that published data indicate that Ub deacetylation is a promiscuous/redundant process insofar as various deacetylases appear to be involved[19].

Beyond verification by western blot analysis, two of the interactions were additionally confirmed using recombinantly expressed proteins. For this purpose, we chose the Ub hydrolase UCHL3, since it showed an interesting binding pattern with decreased binding to Ub 6AcK and Ub 11AcK, and NDP52 that was strongly enriched with Ub 6AcK (Fig. 6a, b). As shown in Fig. 6c, recombinant UCHL3 and NDP52 showed enrichment profiles that are highly similar to those obtained by AE-MS, indicating again the reliability of our AE-MS-derived results (Fig. 6a).

Levels of Ub 6AcK were reported to be downregulated by rapamycin-induced autophagy[48], providing a potential link between acetylation of Ub at K6 and autophagy. Furthermore, NDP52 binds Ub via its C-terminal zinc finger 2 (ZF2)[12] and is involved in xenophagy of invading pathogenic bacteria that are predominantly decorated by Ub chains linked via the α-amino group of the initial methionine (M1)[7,12]. Therefore, we took a closer look at the interaction of NDP52 with Ub 6AcK in comparison to M1-linked Ub dimers (diUb). As expected, we observed ZF2-dependent binding of NDP52 to both Ub 6AcK and M1-linked diUb (Fig. 6d, e). Interestingly, NDP52 bound to diUb and Ub 6AcK with similar efficiency, irrespective of whether diUb/Ub 6AcK or NDP52 were used as baits. Since K6 is part of the ZF2 binding interface of Ub[12], we examined the contact interface between Ub and ZF2 in the

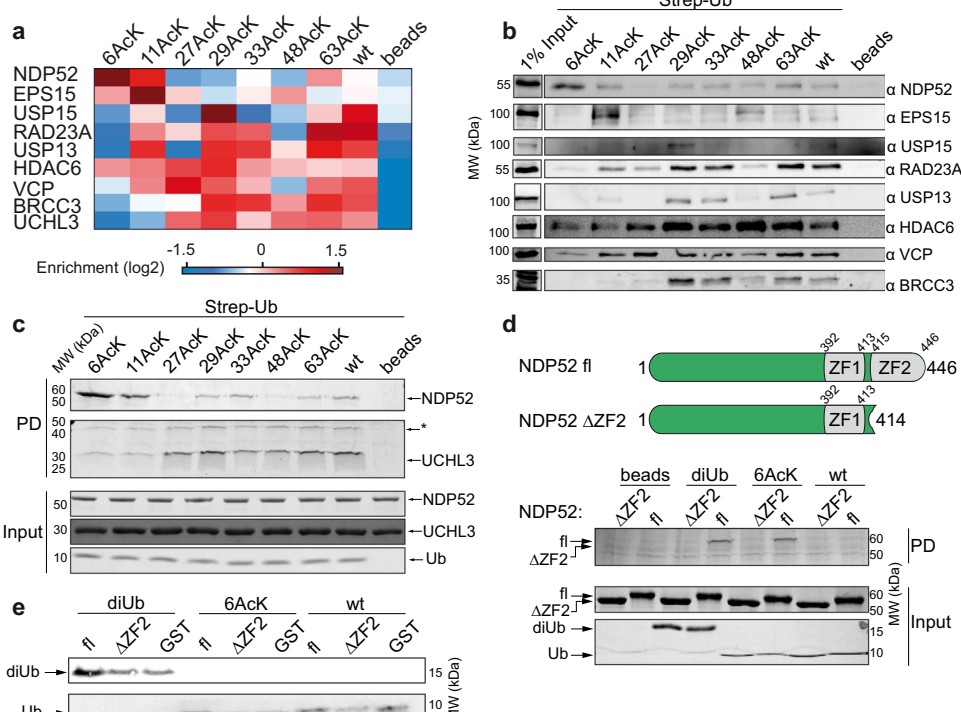

**Fig. 6 | NDP52 binds selectively to Ub 6AcK. a** Selected interactors of acetylated Ub variants with a distinct binding pattern. Red indicates enrichment, whereas blue indicates lack of enrichment. **b** Confirmation of interactions by western blot analysis. 20% of the elution fraction of the affinity enrichment were subjected to western blot analysis with antibodies specific for the indicated proteins. Input represents 1% of the HEK293T cell extract used for affinity enrichment. **c** Verification of interactions by in vitro binding assays using recombinant NDP52 and UCHL3 and the bait molecules indicated. Binding reactions were analyzed by SDS-PAGE followed by Coomassie blue staining (PD). 10% of the proteins used in the binding assay are shown in the lower panel (Input). Asterisk indicates the running position of an UCHL3-Ub adduct that was recently described[80]. **d** In vitro binding assay using Strep-tagged versions of nonmodified Ub (wt), Ub acetylated at K6

(6AcK), M1-linked Ub dimers (diUb), and empty beads as affinity matrices and recombinant full-length NDP52 (fl) or a truncation variant of NDP52 devoid of the C-terminal Ub-binding zinc finger (ΔZF2). Binding reactions were analyzed by SDS-PAGE followed by Coomassie blue staining (PD). 10% of the proteins used in the binding assay are shown in the lower panel (Input). A schematic of full-length NDP52 and NDP52 ΔZF2 is shown at the top. **e** In vitro binding assays with GST fusion proteins of full-length NDP52 (fl) and NDP52 ΔZF2 with the Ub variants indicated. GST was used as control. Proteins bound to the different GST proteins were analyzed by western blot using an antibody directed against Ub. **b–e** The experiments shown are representative of two independent experiments. Source data are provided as a Source Data file.

respective crystal structure (PDB ID 4XKL)[12] and realized that the side chain of K6 points towards a hydrophobic pocket in NDP52 consisting mainly of F429, I436, and F437 (Supplementary Fig. 6). Hence, we hypothesized that either the increase in hydrophobicity upon K6 acetylation or structural changes induced by K6 acetylation results in stronger binding to NDP52. We tested the first hypothesis by replacing K6 with arginine and assayed the ability of Ub 6R to bind NDP52 in comparison to nonmodified Ub and Ub 6AcK (Supplementary Fig. 6). We observed that Ub 6R binds less efficiently than Ub 6AcK to NDP52 (~2-fold difference), while it binds more efficiently than nonmodified Ub (~4-fold difference). This suggests that both an increase in hydrophobicity and structural changes contribute to the enhanced interaction between NDP52 and Ub 6AcK.

## p300 and HDAC6 mediate Ub acetylation and deacetylation, respectively

While the above results provide intimate insights into the structural effects of site-specific Ub acetylation and into the interaction landscape of the various Ub AcK variants, they do not unravel the processes or enzymes involved in determining the acetylation status of Ub. Treatment of human cells with either deacetylase or acetyltransferase inhibitors affects the acetylation status of several lysine residues of Ub[19,21], indicating that, as expected, an interplay between respective enzymes plays a crucial role. Furthermore, a close analysis of the dataset of a recent study investigating the

CBP/p300-dependent acetylome in mouse embryo fibroblasts indicates that knockout of the genes encoding the acetyltransferases p300 and CBP results in reduced levels of acetylated Ub[21] (Supplementary Fig. 7). Hence, we investigated whether p300 can acetylate Ub in vitro by incubating free monomeric Ub with p300 in the absence and presence of the cofactor acetyl-CoA (Fig. 7a). This showed that in the presence of both p300 and acetyl-CoA Ub was acetylated at K48. Since Ub acetylation was initially identified by analyzing Ub chains[19], we next determined if p300 can also acetylate M1-linked diUb. Indeed, we even observed increased acetylation of K48 and, in addition, acetylation at K11 (Fig. 7b). Remarkably, our in vitro results are in good agreement with the aforementioned in vivo data investigating the CBP/p300-dependent acetylome[21]. The levels of Ub acetylated at K48 and K11 decreased upon CBP/p300 knockout or upon treatment with CBP/p300 acetyltransferase inhibitors, while other acetylation sites (K6 and K33) were not affected (Supplementary Fig. 7). Thus, we conclude that CBP/p300 are bona fide Ub acetyltransferases with a preference for certain lysine residues and possibly distinct Ub chains.

In our AE-MS approach, HDAC6 interacted with all acetylated Ub variants, indicating that it acts as a general Ub deacetylase. To prove this possibility, we incubated recombinant HDAC6 with the acetylated Ub variants and analyzed the reaction products by ESI-MS. This showed that, indeed, HDAC6 is capable of deacetylating all of the acetylated Ub variants with similar efficiency (Fig. 7c), while

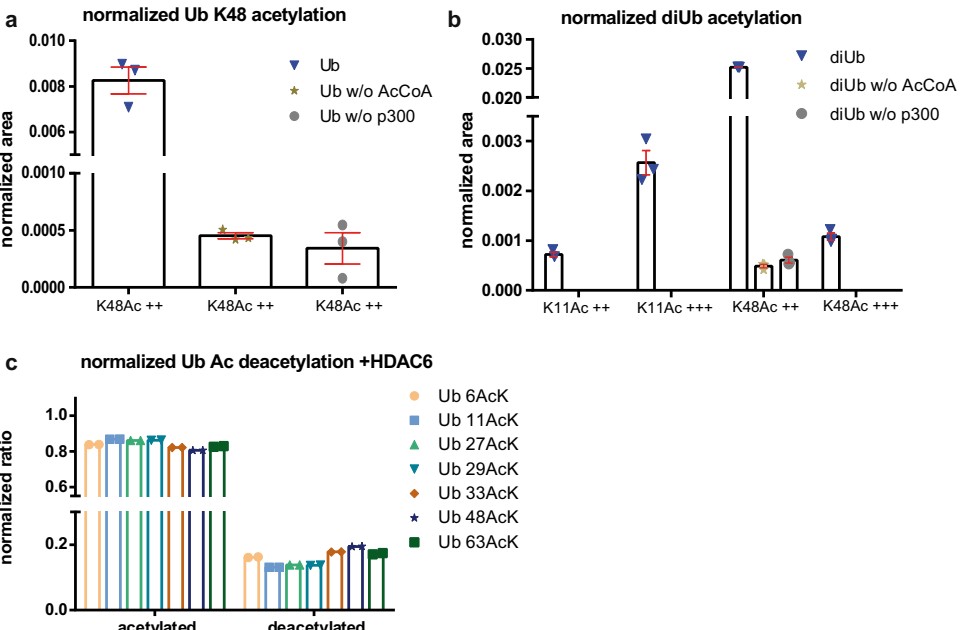

**Fig. 7 | Acetylation and deacetylation of Ub / Ub AcK variants by p300 and HDAC6, respectively. a** Monomeric Ub is acetylated at K48 by p300 in presence of acetyl-CoA (Ub), while only a background signal can be observed in the absence of either acetyl-CoA (Ub w/o AcCoA) or p300 (Ub w/o p300). **b** M1-linked Ub dimers are acetylated at K11 and K48 by p300 in presence of acetyl-CoA (diUb), while in its absence (diUb w/o AcCoA) or in the absence of p300 (diUb w/o p300) only background signals are observed. **a, b** Ub acetylation was normalized against a highly abundant C-terminal peptide of Ub (aa 64–73). ++ and +++ indicate doubly and triply protonated charge states of the detected peptide, respectively. For quantification, the respective product ions (MS2-level) of the indicated precursors were used. Error bars indicate SEM of three independent measurements. **c** HDAC6 deacetylates the various Ub AcK variants with similar efficiency. Ub AcK variants were incubated with HDAC6 and reaction products were analyzed by intact protein MS. Peak areas of the respective acetylated and deacetylated Ub variants were used for quantification of deacetylation and normalized to the total Ub peak area. Data were obtained in two independent experiments. Source data are provided as a Source Data file.

deacetylation was not observed in the absence of HDAC6 (Supplementary Fig. 7). This classifies HDAC6 as a bona fide Ub deacetylase.

## Discussion

With the discoveries that Ub is subject to post-translational modifications, including phosphorylation and acetylation[17], and that phosphorylation of Ub at S65 has a significant impact on PARKIN-mediated mitophagy[15,16], a new facet has been added to the study of Ub structure and function. Here, we show that acetylation affects the structure of Ub in a position-specific manner. In consequence, different acetylated Ub variants have distinct protein-protein interaction properties providing intimate insights into their potential functions.

So far, site-specifically acetylated Ub variants have mainly been characterized for their ability to be used by the enzymes of the Ub-conjugation machinery[18,19]. Our results obtained with different E3 ligases (HDM2, RLIM, and E6AP) confirm previous results showing that acetylation of Ub at K6 or K48 interferes with ubiquitylation/Ub chain formation[19] and extend these by the observation that acetylation of K11 and K27 also interferes with E3-mediated ubiquitylation. Furthermore, this effect is mainly due to inefficient usage of the acetylated Ub variants by the E3 enzymes, or the E2/E3 complex in the case of RING type E3 ligases, although it should be noted that in addition, some of the variants may be less efficiently activated by the E1 enzyme[49]. Notably, with the exception of E6AP and the Ub 48AcK variant, the inefficient usage of the acetylated variants does not correlate with the lysine residues used by the E3s for Ub chain formation. In fact, our mutational analysis indicates that the loss of the positive charge at the respective position is the main determinant for inefficient usage. Moreover, we speculated that acetylation affects the structure of Ub, and thereby its protein interaction surface, in a position-specific manner. Indeed, our profound NMR analyses show that the performance of the different Ub AcK variants in ubiquitylation assays correlates with structural changes

in the I44 and I36 hydrophobic patches, which represent an interaction surface for many Ub-binding proteins and play an important role in E6AP-mediated ubiquitylation[34–37]. Strikingly, the patterns of the CSP mappings of the acetylated Ub variants are highly similar to those of the proximal Ub moiety of respectively linked Ub dimers, except for K6-linked and K48-linked dimers, where the proximal moiety is affected by interdomain interactions with the distal moiety[47]. Nonetheless, for the other linkages (K11, K27, K29, K33, and K63), this indicates that the covalent attachment of a small chemical group via an amide bond has a comparable effect on Ub structure as the attachment of Ub itself. In agreement with this assumption, Boc modification of lysine side chains affects Ub structure in a similar manner[41]. Thus, it is intriguing to speculate that acetylation of Ub may affect the protein interaction properties of Ub in a similar manner as Ub dimer formation, i.e., in those cases, where the interaction surface is mainly provided by the proximal moiety.

By a combination of immunoprecipitation and quantitative high-resolution mass spectrometry, it was shown that under normal growth conditions, levels of acetylated Ub are rather low in comparison to the total level of Ub[19]. Thus, it remains to be seen, if the inefficient usage of Ub AcK variants for Ub chain formation is of physiological relevance and, if so, under which circumstances. In contrast to such "loss-of-function" effects, "gain-of-function" effects are less or even not dependent on the actual abundance of a protein of interest. Along this line and in view of the observation that the effect of acetylation on Ub structure depends on which lysine residue is acetylated, it seems likely that the functional consequences of Ub acetylation are at least in part mediated by proteins that selectively interact with distinct acetylated Ub variants. In support of this hypothesis, our AE-MS approach revealed overlapping, but distinct interactomes of the acetylated Ub variants. GO term analysis, for instance, shows that the 19 S regulatory particle (RP) of the 26 S proteasome is significantly enriched by Ub

48AcK. While we do not know, whether this interaction is mediated by the known Ub-binding proteins of 19 S RP or by other subunits, it may indicate that in certain cases, the attachment of a single Ub 48AcK moiety suffices to target a protein for proteasome-mediated degradation[46]. Another interesting candidate is NDP52, which shows a remarkable selectivity for Ub 6AcK and has been associated with autophagy/xenophagy[7,12]. By analyzing a recent acetylome study performed upon rapamycin-induced autophagy[48], we found that Ub 6AcK levels are downregulated by ~20%. At first glance, this may appear to be contradictory to the notion that Ub 6AcK plays a role in autophagy. However, considering that NDP52 recognizes ubiquitylated proteins that in turn end up in the lumen of the autophagosome and are eventually degraded in the lysosome[50–52], a decrease of Ub 6AcK levels is actually expected.

Besides the identification of proteins that potentially mediate the consequences of Ub acetylation, we did not identify proteins that are known to function as deacetylase or acetyltransferase, with the exception of the deacetylase HDAC6. This was not unexpected since HDAC6 exhibits one of the strongest known binding affinities for free monomeric Ub[53]. Furthermore, we show that HDAC6 can deacetylate Ub AcK variants in vitro irrespective of the actual position of acetylation. Thus, it is tempting to speculate that HDAC6 is generally involved in regulating Ub AcK levels in cells. With regard to Ub acetylation, respective enzymes show, in general, relative weak binding affinities for their substrates[54–56], providing a reasonable explanation for their absence in our AE-MS approach. However, data obtained in cells[21] and our in vitro data provide strong evidence that CBP/p300 play a considerable role in the acetylation of Ub at K48 and K11. Notably, acetylation of K48 and K11 correlates well with the observation that CBP/p300 preferably acetylate lysine residues with a glycine at position −1[48,57]. In future studies, it will be important to identify further enzymes as well as stress stimuli regulating or inducing Ub acetylation at distinct lysine residues.

To delineate the physiological consequences of Ub acetylation, it will be necessary to establish systems that enable manipulating levels of Ub AcK variants within cells. The replacement of respective lysine residues by arginine or glutamine is commonly used to obtain insights into the consequences of acetylation of a protein of interest. However, previous reports[31,32] and our results obtained with Ub 11AcK and Ub 11Q indicate that, at least in certain cases, charge neutralization does not suffice to mimic the effect of lysine acetylation and that, in addition, steric effects have to be considered. Thus, results obtained by mutational analysis may only partially reflect the actual effects of lysine acetylation. An attractive alternative is to apply the genetic expansion method in eukaryotic cells, including yeast as an easy-to-handle model organism that also harbors acetylated Ub variants[58]. While this is doable, a potential caveat is that upon incorporation into Ub (or other proteins), acetyllysine residues may be rapidly hydrolyzed by the action of deacetylases. To circumvent this, non-hydrolyzable acetyllysine analogs may prove useful, as long as they faithfully mimic the structural effects of lysine acetylation.

## Methods
### Plasmids
Codon-optimized cDNA encoding Ub with a C-terminal $His_6$-tag was inserted into the pGEX-2TK backbone replacing the GST cDNA to yield pKS Ub-His. cDNAs of Ub variants, in which the amber stop codon (TAG), a glutamine, an alanine, or an arginine codon replaced the respective lysine codon were generated via site-directed mutagenesis.

The cDNA encoding the aminoacyl-tRNA synthetase for acetyllysine (AcK-RS) derived from *Methanomethylophilus alvus* was kindly provided by Dr. K. Lang (LMU Munich, now ETH Zürich) and was inserted into the pRSFDuet vector (Merck Millipore). The NDP52 cDNA was kindly provided by Dr. Lifeng Pan (Shanghai Institute of Organic Chemistry).

For insertion of the N-terminal Strep-tag II in front of the Ub variants and deletion of the ZF2 domain of NDP52 (NDP52 ΔZF2), the Q5 Site-Directed Mutagenesis Kit (New England Biolabs) was used. Furthermore, for the generation of N-terminal Strep-II tagged diUb, an additional N-terminal Strep-II tagged Ub sequence with a G76 to V mutation was inserted in front of the Ub-His sequence in the pKS Ub-His vector.

cDNAs encoding the UBA domain of Ubiquilin-2, NDP52, and NDP52 ΔZF2 (deletion of the C-terminal 32 amino acids) were cloned into pGEX-2TK with a TEV cleavage site in front of the respective cDNA.

All oligonucleotides used for cloning are listed in Supplementary Table 1.

### Bacterial expression and protein purification
Human $His_6$-tagged UBA1[16], N-terminal $His_6$-tagged UCHL3[16], C-terminal $His_6$-tagged UbcH7[59], C-terminal $His_6$-tagged UbcH5b[16], N-terminal $His_6$-tagged E6AP[59], GST-RLIM-RING[59], GST-ULP1[60], and $His_6$-tagged $TEV_{opt}$[61] were expressed and purified[16,59–61].

Expression of the various GST fusion proteins was performed in *E. coli* BL21 (DE3) Rosetta. For the generation of a GST-SUMO fusion protein of HA-tagged HDM2, cells were grown at 30 °C in LB medium, containing 100 μg/l carbenicillin and 34 μg/l chloramphenicol, until an $OD_{600} = 0.7$ was reached. After harvesting the cells, resuspending them in 42 °C warm 2×YT medium and incubating them for 30 min at 42 °C, the culture was diluted to an $OD_{600} = 0.3$ with ice-cold 2×YT medium and incubated at 16 °C until an $OD_{600} = 0.6$ was reached. Gene expression was induced by the addition of 10 μM IPTG (final conc.), and cells were incubated at 16 °C. After 48 h, cells were harvested and lysed by sonication in 1× PBS, 0.1% Triton X-100, 1 mM DTT supplemented with protease inhibitors (1 mg/ml aprotinin/leupeptin, 1 mg/ml Pefabloc). After centrifugation, GST-SUMO-HA-HDM2 was purified via affinity chromatography (Glutathione Sepharose 4B beads, GE Healthcare), and HA-HDM2 was liberated by addition of GST-ULP1 for 7 h at 4 °C. Eluted HA-HDM2 was dialyzed against 150 mM NaCl, 50 mM Tris-HCl pH 7.5, 5% glycerol, 1 mM DTT, 0.1% NP40.

For the GST fusion protein of the UBA domain of Ubiquilin-2, cells were grown in LB medium, supplemented with 100 μg/l carbenicillin/34 μg/l chloramphenicol. Gene expression was induced at $OD_{600} = 0.7$ by 1 mM IPTG (final conc.) and grown overnight at 25 °C. Cells were harvested and lysed in 20 mM Tris-HCl pH 7.6, 100 mM NaCl, 0.02% Triton X-100, 1 mM DTT, 1 mg/ml aprotinin/leupeptin and 1 mg/ml Pefabloc. The GST fusion protein was purified via Glutathione Sepharose 4B beads (GE Healthcare), and elution of the free UBA domain was achieved by the addition of His-TEV for 16 h at 4 °C. The eluate was concentrated by Amicon Ultra and was further purified via size exclusion chromatography (Superdex S75 16/600 column, GE Healthcare) in 1× PBS supplemented with 300 mM NaCl. Pure fractions were pooled and dialyzed against 2.85 mM $NaH_2PO_4$ pH 6.8 and subsequently concentrated by SpeedVac. UBA domain concentration was determined via absorbance at 205 nm.

For full-length NDP52 and NDP52 ΔZF2, cells were grown in LB medium, supplemented with 100 μg/l carbenicillin/ 34 μg/l chloramphenicol, until $OD_{600} = 0.7$ was reached. Upon induction of gene expression by 1 mM IPTG (final conc.), cells were grown overnight at 25 °C. Cells were harvested and lysed by sonication in 50 mM Tris-HCl pH 8, 300 mM NaCl, 1 mM DTT, 0.01% Triton X-100 supplemented with protease inhibitors (1 mg/ml aprotinin/leupeptin, 1 mg/ml Pefabloc). After centrifugation, the GST-tagged proteins were purified via Glutathione Sepharose 4B beads (GE Healthcare) and cleaved by addition of His-TEV for 16 h at 4 °C. Upon centrifugation, NDP52 and NDP52 ΔZF2 in the supernatant were further purified by 1 ml HiTrap Q HP (GE Healthcare) with a gradient from 25 mM Tris-HCl pH 7.5, 1 mM DTT, 50 mM NaCl to 25 mM Tris-HCl pH 7.5, 1 mM DTT, 1 M NaCl. NDP52/NDP52 ΔZF2 containing fractions were pooled and dialyzed against 20 mM Tris-HCl pH 7.5, 100 mM NaCl, 1 mM DTT.

## Generation of acetylated Ub and nonmodified Ub

For the generation of site-specifically acetylated Ub (with and without an N-terminal Strep-II tag), the respective pKS vector was co-transformed with pRSFDuet harboring the AcK-RS/tRNA pair for amber codon suppression into *E. coli* B834 (DE3) cells. Cells were grown at 37 °C in 100 ml LB medium containing 100 μg/l carbenicillin/34 μg/l kanamycin till $OD_{600} = 1$ was reached. Then, cultures were diluted to an $OD_{600} = 0.1$ with minimal media devoid of phenylalanine and further incubated. At an $OD_{600} = 0.4$, 10 mM acetyllysine (AcK, abcr) and 20 mM nicotinamide (Sigma–Aldrich) were added. Gene expression was induced at $OD_{600} = 0.7$ by the addition of 1 mM IPTG (final conc.). After incubation at 25 °C for 16 h, cells were harvested by centrifugation, resuspended in lysis buffer (50 mM Tris-HCl pH 7.4, 150 mM NaCl, 30 mM imidazole, 1% Triton X-100, 1 mg/ml aprotinin/leupeptin, 1 mg/ml Pefabloc) and lysed by sonication. The lysate was cleared, and acetylated Ub-His was purified using a 5 ml HisTrap HP (Äkta pure 25, GE Healthcare) with a step from 30 mM to 242 mM imidazole in 50 mM Tris-HCl pH 7.4, 150 mM NaCl for elution. Protein-containing fractions were pooled, UCHL3 was added to a final concentration of 75 μg/ml to remove the C-terminal His-tag, and the samples were incubated for 4 h. After cleavage, Ub AcK variants were dialyzed against 25 mM acetate pH 4.5 overnight at 4 °C. Ub AcK samples were further purified via a 1 ml HiTrap SP HP (GE Healthcare) with a linear gradient from 25 mM acetate pH 4.5 to 25 mM acetate 4.5, 1 M NaCl. Ub AcK-containing fractions were pooled and dialyzed against 25 mM Tris-HCl pH 7.5, 50 mM NaCl. The concentration was determined with the BCA Protein Assay Kit (Thermo) and commercial Ub (Sigma–Aldrich) as standard. The correct masses of the Ub AcK variants and the site of incorporation of AcK were confirmed by ESI-MS and LC-MS/MS, respectively.

Expression and purification of nonmodified Ub and the Ub variants containing Q, A, or R instead of the respective K were performed similarly to the acetylated Ub variants, except that AcK and nicotinamide were not added to the respective expression cultures. Furthermore, N-terminal Strep-II tagged M1-linked diUb was expressed in *E. coli* DH5α and purified as described above.

## In vitro autoubiquitylation assay

For in vitro autoubiquitylation, 100 nM UBA1, 0.5 μM UbcH5b, and 0.55 μM HDM2, 0.25 μM E6AP or 0.75 μM GST-RLIM RING were incubated in 25 mM Tris-HCl pH 7.5, 50 mM NaCl, 1 mM DTT, 2 mM ATP, and 2 mM $MgCl_2$. The reaction was started by the addition of 7.8 μM (final conc.) of the respective Ub variants and incubated for the times indicated at 37 °C. Reactions were stopped by the addition of 5× reducing SDS loading buffer. Entire reaction mixtures were electrophoresed in 12.5% SDS-PA gels followed by Coomassie blue staining.

For in vitro autoubiquitylation assays with UbcH7, reactions were performed similarly except that instead of UbcH5b, 0.5 μM UbcH7 was used, and the reactions were performed with E6AP only.

Autoubiquitylation experiments were performed at least three times with similar results.

## Generation of isotopically labeled proteins for NMR analysis

For the preparation of $^{13}C$–$^{15}N$-labeled proteins, *E. coli* harboring the respective expression construct was grown in LB medium at 37 °C until $OD_{600} = 1$. This culture was used to inoculate $^{15}N/^{13}C$ M9 medium (33.7 mM $Na_2HPO_4$, 22 mM $KH_2PO_4$, 8.55 mM NaCl, 9.35 mM $^{15}NH_4Cl$, 1 mM $MgSO_4$, 0.3 mM $CaCl_2$, 134 μM EDTA, 31 μM $FeCl_3$, 6.2 mM $ZnCl_2$, 0.76 μM $CuCl_2$, 0.42 μM $CoCl_2$, 1.62 μM $H_3BO_3$, 0.081 μM $MnCl_2$, 1 mg/l biotin, 1 mg/l thiamine, 0.2% $^{13}C$ glucose (w/v)). Upon growth overnight at 37 °C, 100 ml of the culture were diluted with 900 ml of pre-warmed $^{15}N/^{13}C$ M9 medium. Cells were grown at 37 °C until $OD_{600} = 0.4$ was reached, and then 10 mM AcK (abcr) and 20 mM nicotinamide (Sigma–Aldrich) (final conc.) were added. Gene expression was

induced at $OD_{600} = 0.7$ by the addition of 1 mM IPTG (final conc.). After incubation at 25 °C for 16 h, cells were harvested by centrifugation.

For $^{15}N$ labeled proteins, the procedure was as described for doubly labeled proteins, but the M9 medium contained 0.4% (w/v) glucose instead of $^{13}C$ glucose.

Purification of all isotopically labeled proteins was performed as described above for their unlabeled counterparts.

## Acquisition of NMR data

NMR experiments were performed at *T* = 298 K on a Bruker Avance III 600 MHz spectrometer equipped with a TCI cryoprobe or an Avance Neo 800 MHz spectrometer equipped with either TCI triple or QCI cryoprobe. Datasets were processed using NMRPipe[62] and analyzed by NMRView[63]. Samples were prepared in 20 mM $NaH_2PO_4$ (pH 6.8) with 5% (v/v) $D_2O$. All measurements were performed with singly $^{15}N$ labeled proteins except for Ub 27AcK, which was $^{13}C/^{15}N$ labeled. The concentrations ranged between 20 and 90 μM.

Two-dimensional $^1H$–$^{15}N$ HSQC spectra were collected for all Ub AcK variants and for Ub variants where either K11 or K48 were replaced by Q, A or R. Additionally, $^1H$–$^{15}N$ HSQC spectra were obtained for nonmodified Ub. The spectra were recorded with 1024 and 256 increments in the direct $^1H$ and indirect $^{15}N$ dimension, respectively, using 8–32 transients depending on the concentration and magnetic field strength. WATERGATE pulse sequence and presaturation were applied for efficient solvent suppression.

## Assignment procedure of NMR spectra

Whenever unambiguously applicable, the assignment of backbone amide cross peaks in the two-dimensional spectra of the Ub variants was accomplished by visual estimate. For this purpose, we used properly assigned spectra of Ub where one of the corresponding K residues was replaced by C[39,64]. Additionally, spectra from samples where C was modified by propargyl acrylate and from samples of artificially triazole-linked Ub dimers where only the proximal subunit is isotopically labeled served as the basis for the assignment[39,64]. In most cases, the procedure was successful by searching for the next neighboring peaks and following peak trajectories.

For Ub 48AcK, the remaining ambiguities were solved by a three-dimensional $^1H$–$^{15}N$ NOESY-HSQC spectrum on a sample with a concentration of ~150 μM. The corresponding spectrum was recorded with 8 transients and 1024, 256, and 64 increments in the $^1H$, $^{15}N$, and indirect $^1H$ dimension, respectively. The mixing time was set to 120 ms.

As the $^1H$–$^{15}N$ HSQC spectrum of Ub 27AcK revealed extraordinary strong perturbations, the assignment was completed by means of TROSY-based three-dimensional triple resonance experiments using a sample with a concentration of ~876 μM. We conducted 16 transients of an HNCA experiment with 1024, 80, and 72 increments, 8 transients of an HNCO experiment with 1024, 80, and 72 increments, and 8 transients of an HN(CO)CACB experiment with 1024, 72, and 80 increments in the $^1H$, $^{15}N$, and $^{13}C$ dimension, respectively.

## Calculation of chemical shift perturbations

Weighted chemical shift perturbations between the different Ub variants and nonmodified Ub were calculated according to the following equation[65]:

$$\triangle\omega = \sqrt{\frac{(\triangle^1H)^2 + \frac{1}{25}(\triangle^{15}N)^2}{2}}, \tag{1}$$

where $\Delta^1H$ is the peak shift in proton and $\Delta^{15}N$ is the peak shift in nitrogen dimension, respectively, between two corresponding peaks.

## NMR titration experiments

NMR interaction studies were conducted at a magnetic field strength of 18.8 T at *T* = 298 K, and all samples were prepared in the same buffer

(20 mM $NaH_2PO_4$ pH 6.8 with 5% (v/v) $D_2O$). The unlabeled Ubiquilin-2 UBA domain served as the ligand and was added stepwise to $^{15}N$ isotopically labeled samples of Ub 11AcK, Ub 11Q, Ub 48AcK, and Ub 48Q using starting concentrations of $c = 33\,\mu M$ for Ub 11AcK and $c = 157\,\mu M$ for the other Ub variants. The stock concentration of the UBA domain was adjusted to $c = 2.031\,mM$ for the titration with Ub 48AcK and $c = 803\,\mu M$ for the other experiments. Titrations were performed in 10 steps for Ub 48AcK, and Ub 48Q and in 7 steps for Ub 11AcK and Ub 11Q. For nonmodified Ub, starting concentrations of $c = 33\,\mu M$ and 7 titration steps or $c = 157\,\mu M$ and 10 titration steps were applied as indicated. At each step, a $^1H$–$^{15}N$ HSQC spectrum was acquired with 1024 and 256 increments in the direct $^1H$ and indirect $^{15}N$ dimension, respectively, using 8–32 transients. Changes in chemical shifts compared to the corresponding spectrum at starting conditions were calculated according to Eq. (1) for each backbone amide resonance.

For quantification of the binding affinity, six residues were selected (T14, L43, F45, E51, L67, and L71) that show significant CSP values in all titration experiments but do not experience intensity loss by more than 90% due to line broadening during the titration course. Assuming a 1:1 binding stoichiometry, dissociation constants ($K_d$) were computed from the following equation by applying a global fitting procedure[66]:

$$\triangle\omega^{obs} = \triangle\omega^{max} \frac{[P]_t + [L]_t + K_d - \sqrt{([P]_t + [L]_t + K_d)^2 - 4[P]_t[L]_t}}{2[P]_t}, \quad (2)$$

where $\Delta\omega^{obs}$ is the CSP value observed at a distinct titration point, $[P]_t$ and $[L]_t$ are the corresponding concentrations of the Ub variant and the UBA domain, respectively, and $\Delta\omega^{max}$ is the maximum CSP value assumed to be reached at saturation.

## Molecular dynamics simulations and analysis

All simulations were carried out using the GROMACS software suite (version 2018 and 2020)[67,68] applying the GROMOS96 54A7[69] force field. Starting structures for nonmodified Ub were prepared from PDB ID 1UBQ[70]. The variants 11A, 11Q, and 11R were generated by replacing the respective residue with chimera[71] and selecting the best fitting rotamer from the Drunback rotamer library[72]. For the acetylated variant Ub 11AcK, the structure file was created with the Vienna-PTM server[73,74] using PDB ID 1UBQ[70] as input. The termini and side chains were defined as charged.

Periodic boundary conditions were applied with a dodecahedral box and a minimum distance of 1.0 nm between the protein and the box wall. The protein structure was energy minimized in vacuum (50,000 steps, steepest descent), solvated with SPC/E water, neutralized with $Na^+$ ions, and energy minimized (50,000 steps). Three equilibration steps of 100 ps each were carried out at NVT (300 K) conditions and at NPT (300 K, 1 bar) conditions with position restraints, and finally at NPT (1 bar) conditions without position restraints.

The Leap frog integrator was used with a time-step of 2 fs, with bond constraints on all bonds using the LINCS algorithm. The temperature was maintained at 300 K by the velocity rescaling method[75] with a period of temperature fluctuations at equilibrium set to 0.1 ps. Constant pressure was maintained at 1 atm using isotropic Parrinello-Rahman pressure coupling with a pressure relaxation time of 2 ps. The Verlet cutoff scheme was used for short-range Van der Waals interactions with a cutoff distance of 1.4 nm. Long-range coulomb interactions were calculated by particle mesh Ewald (PME)[76] method with the same cutoff. The neighbor list was updated every 10 steps.

Three independent production simulations at NPT conditions of 1000 ns each were performed for each structural model.

To monitor the influence of the different side chains at position 11 (acetylated lysine 11AcK, arginine variant 11R, glutamine variant 11Q, and alanine variant 11A) on the local environment, we calculated two

descriptors: The pairwise minimum distance of the residues 11 and 34 in the respective Ub variants and nonmodified Ub; and a matrix representing the difference between the pairwise average minimum distance between all residues in the variants with respect to the same distance in nonmodified Ub. The matrices were calculated with the mdmat module of GROMACS for the entire protein with a cutoff of 1 nm.

## Cell culture and extract preparation

HEK293T cells (ATCC) were cultured in Dulbecco's Modified Eagle's Medium (DMEM, Thermo Fisher) with 10% fetal bovine serum (FBS, Sigma).

Cell pellets were resuspended in cell lysis buffer (1× PBS, 2 mM $MgCl_2$, 1 mM DTT, supplemented with 1 mg/ml aprotinin/leupeptin, 1 mg/ml Pefabloc) and subsequently lysed by sonication. After centrifugation (21,000 × $g$, 30 min, 4 °C), the total protein concentration of the supernatant was determined with the BCA Protein Assay Kit (Thermo Fisher) and freshly dissolved BSA as standard.

## Affinity enrichment

For affinity enrichment, 25 µg of each Strep-Ub variant was incubated with 2.5 mg whole cell extract for 10 min on ice. After washing of 50 µl Strep-Tactin® beads (iba) with three times 200 µl cell lysis buffer, the mixture was added to the beads. Incubation at room temperature for 3 h in an overhead shaker was followed by five washing steps with 200 µl cell lysis buffer each. Bound proteins were eluted by incubating the beads for 5 min on ice with two times 150 µl and one time 100 µl elution buffer (100 mM Tris-HCl pH 8.0, 150 mM NaCl, 1 mM EDTA, 2.5 mM desthiobiotin). The combined eluate was incubated again for 5 min with the beads on ice, before final elution and freeze-drying. Each experiment was performed in quadruplicates.

The freeze-dried eluate was resuspended in 100 µl 8 M urea, reduced with 5 mM TCEP for 30 min at 37 °C and alkylated with 10 mM iodoacetamide for 30 min at room temperature. After dilution to 1 M urea with 50 mM $NH_4HCO_3$, 5 µg trypsin was added, and proteins were digested for 20 h at 37 °C. The digested samples were freeze-dried and kept at −20 °C until they were prepared for MS measurement.

Prior to MS measurement, the tryptic peptides were dissolved in 50 µl 0.1% TFA in $H_2O$, acidified with 10% TFA and desalted using Pierce™ C18 spin tips (Thermo Fisher).

For western blot analysis, affinity enrichment was performed as described above. The combined eluates were concentrated to 50 µl (SpeedVac, Thermo Fisher), 20% of which were applied to SDS-PAGE followed by western blot analysis using antibodies specific for the respective protein (anti-NDP52 (1:1000; #60732, Cell Signaling Technology CST), anti-EPS15 (1:1000; #12460, CST), anti-USP15 (1:1000; #66310, CST), anti-RAD23A (1:1000; #ab108592, abcam), anti-USP13 (1:1000; #ab109264, abcam), anti-HDAC6 (1:1000; #7558, CST), anti-VCP (1:1000; #2649, CST), anti-BRCC3 (1:1000; # ABIN1586883, antibodies-online)), and as secondary antibody anti-rabbit HRP (1:15,000; #111-035-003, Jackson ImmunoResearch).

## Mass spectrometry

Peptide samples were analyzed on a Q-Exactive HF mass spectrometer (Thermo Fisher Scientific, Bremen, Germany) operated with Tune (version 2.9) and interfaced with an Easy-nLC 1200 nanoflow liquid chromatography system (Thermo Scientific, Odense, Denmark). Samples were reconstituted in 0.1% formic acid and loaded onto the C18 analytical column (75 µm × 15 cm). Peptides were resolved at a flow rate of 300 nl/min using a linear gradient of 6–40% solvent B (0.1% formic acid in 80% acetonitrile) over 165 min. Data-dependent acquisition with full scans over a 350–1500 $m/z$ range was carried out using the Orbitrap mass analyzer at a mass resolution of 60,000 at 200 $m/z$, automatic gain control target value of $3e^6$, and max injection time of 60 ms. The 10 most intense precursor ions were selected for further

fragmentation. Only peptides with charge states 2–6 were used, and dynamic exclusion was set to 30 sec. Precursor ions were fragmented using higher-energy collision dissociation (HCD) with normalized collision energy (NCE) set to 28%. Fragment ion spectra were recorded at a resolution of 15,000, automatic gain control target value of $1e^5$, and max. injection time 60 ms. Each of the four biological replicates was measured in technical duplicates.

For label-free quantification, the raw files from LC-MS/MS measurements were analyzed using MaxQuant (version 1.6.8) with default settings and match between runs and label-free quantification (LFQ) enabled. For protein identification, the human reference proteome downloaded from the UniProt database (download date: 2018-02-22) was used. For further data processing, Perseus software (version 1.6.10.50) was used. Identified proteins were filtered for reverse hits and common contaminants. LFQ intensities were log2 transformed. Missing values were imputed from a normal distribution (width = 0.3 and shift = 1.2) based on the assumption that these proteins were just below the detection limit. Only proteins that were detected in at least 6 out of 8 replicates (four biological replicates, each measured as technical duplicate) were retained for further analysis. Significantly enriched proteins were identified by an ANOVA test (FDR = 0.002, $S_0 = 4$) and Z-score normalized. After averaging groups by building the median of all replicates, enriched proteins were clustered by correlation and plotted as heatmaps. Additional annotations (GO annotations, KEGG, Pfam) and identification of enriched terms were done with Perseus.

For the peptide quantification of p300-mediated acetylation of Ub and linear diUb, parallel reaction monitoring (PRM) with an inclusion list added to the targeted MS2 mode on a Q-Exactive HF was used. Obtained PRM data were analyzed via Skyline software 21.1.0.146. For this, MS1-filtering isotope precursor ion peak was set to 3, and for peptide quantification MS2-level was used. Obtained values were plotted with GraphPad Prism 6.01.

For intact protein mass measurements, a micrOTOF II (Bruker) interfaced with an Agilent 1260 Infinity II liquid chromatography system was used. The proteins were loaded onto the analytical column (Nucleodur 300-5 C4 ec, EC 150/4, Machery-Nagel) and resolved at a flow rate of 300 μl/min using a linear gradient from 5 to 100% solvent B (0.1% formic acid in acetonitrile) over 15 min. The electrospray ion source was operated in positive ionization mode, scan range 650–2100 $m/z$ at 1 Hz acquisition rate, and 2× rolling average was used. Compass DataAnalysis software (Bruker) was used for re-calibration and spectra deconvolution.

### In vitro binding assay with recombinant proteins
For in vitro coprecipitation assays, 5 μM of the respective Strep-Ub variant were incubated 10 min on ice in binding buffer (1× PBS, 10 mM MgCl₂, 5 mM DTT) with 5 μM of recombinant UCHL3 or NDP52 variants prior to the addition of 5 μl Strep-Tactin® beads (iba) pre-equilibrated in binding buffer. The mixtures were incubated for 3 h at room temperature in an overhead shaker, followed by three washing steps with 100 μl binding buffer each. Bound proteins were eluted twice by incubating beads with 30 μl elution buffer (200 mM Tris-HCl pH 8.0, 300 mM NaCl, 4 mM EDTA, 10 mM desthiobiotin). Eluates were combined and electrophoresed on 10% SDS-PA gels (NDP52 variants) or 12.5% SDS-PA gels (UCHL3), followed by Coomassie blue staining. Coprecipitation assays with GST-NDP52/GST-NDP52 ΔZF2 were performed as described for Strep-Ub except that 5 μM of GST or GST fusion proteins were used and 10 μM of the respective Ub variants. Furthermore, 20 μL Glutathione Sepharose 4B beads (GE Healthcare) were used, and elution was performed with 35 mM glutathione in 1× PBS pH 7.2. Eluates were electrophoresed in 15% SDS-PA gels, and Ub variants were detected by western blot analysis using an anti-Ub antibody (1:1,000; #07-375, Merck Millipore) and as a secondary antibody anti-rabbit HRP (1:15,000; #111-035-003, Jackson ImmunoResearch).

For determining the relative binding affinity of NDP52 to Ub variants, an enzyme-linked immunosorbent assay (ELISA)-like approach was used[77]. In brief, 30 wells of a 96-well plate were coated with a multichannel pipette with 100 μl of the respective Ub variant (100 ng/μl) in 34 mM Na₂CO₃, 100 mM NaHCO₃ buffer. After incubation overnight, the wells were washed three times with 200 μl buffer A (1× PBS, 0.05% Tween 20) and blocked with 5% BSA in 1× PBS for 2 h at room temperature. After washing three times with buffer A, 100 μl of GST-NDP52 or GST-NDP52 ΔZF2 in 1× PBS, 0.1% BSA were added at the concentrations indicated (final conc.) followed by incubation for 2 h at room temperature. After washing with buffer A, 100 μl of an anti-GST antibody (1:1,000; #G7781, Sigma–Aldrich) in 1× PBS, 0.1% BSA was added to each well. Upon incubation for 2 h at room temperature, wells were washed with buffer A and 100 μl of an anti-rabbit antibody (1:10,000; #SBA-4050-05, dianova) in 1× PBS, 0.1% BSA were added, followed by incubation for 1 h at room temperature. For detection, 0.5 ml of 25 mM Tetramethylbenzidine (TMB) in 10% Acetone, 90% EtOH with 0.06% H₂O₂ were added to 10 ml of 30 mM potassium citrate pH 4.1. After washing with buffer A, 100 μl of the mixture was added to each well. After 10 min at room temperature, the reaction was stopped by the addition of 50 μl of 2 M H₂SO₄ per well, and the absorbance at 450 nm was determined. For the calculation of relative binding affinities, values obtained with GST-NDP52 ΔZF2 ("background binding") were subtracted from the corresponding values obtained with GST-NDP52.

### p300-mediated acetylation
For in vitro acetylation of Ub, 200 ng of p300 (Active Motif S.A.) were incubated with 5 μM Ub or N-terminally Strep-tagged linear Ub dimers in 50 mM Tris-HCl pH 8, 1 mM DTT, 1 mM EDTA, 20 μM acetyl-CoA (Sigma–Aldrich), 5 mM butyrate. After incubation for 3 h at 37 °C, reaction mixtures were subjected to SDS-PAGE, and proteins were visualized by colloidal Coomassie staining. Bands were excised, and in-gel digest was performed[16]. In brief, gel pieces were destained, reduced with 10 mM DTT, and alkylated with 50 mM iodoacetate. After shrinking and subsequent wetting with trypsin solution (0.0133 μg/μl) for 2 h at 4 °C, in-gel digests were performed overnight at 37 °C. Then, peptides were extracted by the addition of acetonitrile and supernatant, and extracts were combined and lyophilized. Peptides were dissolved and subjected to desalting using Pierce™ C18 spin tips. Mass spectrometric analysis was performed as described above (Mass spectrometry). p300-mediated acetylation was determined in three independent experiments.

### HDAC6-mediated deacetylation
For in vitro deacetylation of Ub, 1 μg of the respective acetylated Ub variant was incubated with 2 μg of recombinant full-length HDAC6 (Active Motif S.A.) in 25 mM Tris-HCl pH 8.0, 50 mM NaCl, 0.02 mM Zn²⁺. After incubation for 3 h at 30 °C, total reaction mixtures (10 μl) were heat-inactivated and diluted with 10 μl UHPLC water, and subsequent intact protein mass measurements with 2 μl sample injection were performed as described above (Mass spectrometry). HDAC6-mediated deacetylation was determined in two independent experiments.

### Reporting summary
Further information on research design is available in the Nature Research Reporting Summary linked to this article.

## Data availability
The data that support this study are available from the corresponding authors upon reasonable request. The NMR solution structure and the crystal structure of ubiquitin, the NMR solution structure of lysine-free ubiquitin, and the crystal structure of NDP52 ZF2 in complex with ubiquitin used in this study are available in the Protein Data Bank under the accession codes 1D3Z, 1UBQ, 2MI8, and

4XKL, respectively. The mass spectrometry proteomics data have been deposited to the ProteomeXchange Consortium (http://proteomecentral.proteomexchange.org) via the PRIDE repository[78] with the dataset identifier PXD028797 for the AE-MS data and PXD028813 for the PRM data of p300-mediated Ub acetylation. Source data are provided with this paper.

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

## Acknowledgements

We thank Andreas Marquardt and Anna Sladewska-Marquardt of the Proteomics Center of the University of Konstanz for assistance with mass spectrometric experiments, Martin Mex and Daniel Hammler for helpful discussions, Franziska Müller and Fabian Offensperger for providing purified proteins, and Silke Büstorf and Nicole Richter-Müller for tech-nical support. We especially thank Dr. David Fushman for providing CSPs of Ub dimers and Dr. Lifeng Pan and Dr. Kathrin Lang for cDNAs of NDP52 and acetyllysyl-tRNA synthetase, respectively. We thank the University of Konstanz for providing excellent NMR infrastructure. This work was supported by the DFG (SFB969, Projects B3 and B9). C.P. and C.G. acknowledge support by the state of Baden-Württemberg (bwHPC) and by the DFG (INST 35/1134-1 FUGG). S.M.K. thanks the doctoral fund of the University of Konstanz and the Konstanz Research School Chemical Biology for financial support. K.S. acknowledges the Zukunftskolleg of the University of Konstanz for a doctoral fellowship. F.S. is grateful for

funding by the DFG Emmy Noether Programme and the DFG Heisenberg Programme (STE 2517/1, STE 2517/5-1).

## Author contributions

S.M.K., T.S., A.M., M.K., and M.S. conceived the study and experimental approach, S.M.K. generated all Ub variants and other proteins. K.S. provided E1, E2, and UCHL3. S.M.K. performed AE-MS, in vitro p300 acetylation MS experiments, autoubiquitylation assays, and interaction studies for AE-MS verifications. T.S. conducted all NMR spectroscopic analyses. C.G. carried out the MD simulations. S.M.K., T.S., J.J., C.G., F.S., C.P., A.M., M.K., and M.S. analyzed the data. S.M.K., T.S., and M.S. wrote the paper. All authors provided critical feedback on the paper.

## Funding

## Competing interests

The authors declare no competing interests.
