## [Peer Review File · Nature Communications]

REVIEWER COMMENTS

Reviewer #1 (Remarks to the Author):

Review for the article „Electrostatic and steric effects underlie acetylation-induced changes in ubiquitin structure and function”

The lab of Martin Scheffner presents a study to elucidate the role of post-translational lysine acetylation for ubiquitin function. Based on quantitative mass-spectrometry-based screens it was discovered that except from K29 all lysines in ubiquitin can be lysine acetylated. Kienle and co-workers present a thorough analysis of all of the six lysine-acetylation sites in ubiquitin and how acetylation affects structure and function. They used an elegant system, namely, genetic code expansion, to prepare site-specifically lysine-acetylated ubiquitin variants. To this end, they analyzed Ub-chain generation by E1 and selected E2 and E3 enzymes. Moreover, they performed NMR studies to analyze the impact of lysine acetylation on Ubiquitin structure and conformational changes that might affect the interplay with other proteins. These studies revealed that acetylation resulted in alterations in the Ub structure mainly by electrostatic effects by neutralization of the positive charge at the lysine side chain. Moreover, steric effects upon acetylation also contribute to the observed alterations in the Ub structure. Moreover, Kienle and co-workers performed pull-down experiments coupled to MS-based experiments with the acetylated ubiquitin variants to identify proteins for which the interaction was modulated in an acetylation-dependent manner. These experiments revealed proteins that interacted dependent on the presence of an acetyl-group at the various lysine side chains, others that were decreased in the interaction. Some interactions were modified with several acetylations in ubiquitin, while others were modulated specifically dependent on the presence of a specific lysine acetylation site. Among these interactors the authors identified DUBs, components of the 19S regulatory particle of the 26S proteasome. Moreover, these analyses revealed that Ub 6AcK binds specifically to the protein NDP52, which is involved in xenophagy of invading bacterial pathogens. Pull-down experiments confirmed this interaction and showed that the interaction depends on the presence of the acetyl-group at K6 in Ub. Finally, the authors did first studies on the regulation of ubiquitin lysine acetylation by lysine acetyltransferases p300/CBP. The authors performed *in vitro* acetylation of ubiquitin with subsequent MS analysis to show that mono-ubiquitin and Met1-linked di-ubiquitin are acetylated at K48 and K11. Overall, the manuscript presents interesting data on the role of lysine-acetylation on ubiquitin structure and function. The experiments performed were all of high quality. However, from my perspective the physiological significance of the findings are not elucidated enough to justify publication in Nature Communication. In the following points I summarize my major points of criticism:

1. The authors used selected E3 ligases to analyze the impact of the lysine-acetylated ubiquitin variants on ubiquitin chain formation. To this end, they selected HDM2, RLIM, and E6AP/UBE3A and analyzed the autoubiquitylation activity using the acetylated variants as substrates. This selection of E3 enzymes seems to be a bit random and only shows that efficiencies for these E3 ligases is affected not/affected.

The authors conclude from their results that acetylation at K6, K11, K27 and K48 interferes with efficient ubiquitin chain formation. From my perspective, it is not possible to draw general conclusions from these results, i.e. how the activities of E1, E2 and of the app. 600 E3 Ub-ligases are affected by ubiquitin acetylation at a specific site. Maybe the authors should focus on specific cellular processes and elucidate the role of individual acetylation events in ubiquitin, including the analysis of the acetylation stoichiometry and the regulation by acetyltransferases (or non-enzymatic acetylation) and by deacetylases.

2. The experiments analyzing the impact of K to Q, K to R and K to A mutants on ubiquitin chain formation by different E3 ligases are only done once and no quantification of the immunoblotting/Coomassie brilliant blue stained SDS-PAGE gels is performed, which makes it really challenging to judge and compare the efficiencies of the reactions.

3. The authors show that for some mutants the K to Q mutant, often used as a mimic of lysine-acetylation cannot mimic lysine acetylation. Moreover, also the K to R mutants, often used to conserve the positive charge to mimic an non-acetylated lysine does influence ubiquitin chain formation. These data show that electrostatic as well as steric effects are exerted by lysine acetylation to regulate ubiquitin function. As an example, the authors state, that for K6, the positive charge is the major driver for observed repressive effect on E3 ligase function. However if looking at the SDS-PAGE gels (Supp. Fig. 2) it is shown that Ub K6R can be used as substrate for E6AP but compared with Ub WT, the efficiency is clearly reduced (after 30 min there is still some mono-ubiquitin visible). This shows that next to charge neutralization also steric effects are essential to recover the full impact of lysine acetylation.

4. The fact that K to Q and K to R mutants are no perfect tools to study lysine acetylation is not new and there are several publications showing that these can be poor tools to study lysine acetylation.

5. The finding that Ub 6AcK increases the binding towards NDP52 is really interesting as this means a "gain-of-function" for post-translational lysine acetylation. This is rarely observed for this modification. However, the interaction is analyzed only qualitatively and more quantitative data for the interaction would be desirable.

6. Data on the regulation of the different Ub acetylation sites are almost completely missing. This is one of the major points of criticism. To show how the sites are acetylated (enzymatically and/or non-enzymatically) and deacetylated (enzymatically deacetylated or not) would be an important criterion to judge the physiological significance of these modifications. The only data the authors show is on interactome of ubiquitin, where HDAC6 is found, and on CBP/p300. P300/CBP is only capable to acetylate K48 and K11 in ubiquitin. How the other lysine side chains are acetylated is not analyzed further. The authors did not analyze any of the other known acetyltransferases and deacetylases.

7. A major point of criticism is the physiological importance of the observed results. If the Ub acetylation has a loss-of-function effect on E3 mediated Ub chain formation etc. it is necessary that the acetylation reached sufficiently high stoichiometries. The authors should perform studies to show under which physiological conditions the acetylation of ubiquitin is increased and which stoichiometries are reached. This is essential to judge the physiological significance of a specific lysine acetylation event. Along that line, the authors do not present a single immunoblotting using an anti-acetyl lysine antibody that shows the acetylation of ubiquitin. Mass-spectrometry based methods are highly sensitive and will also show

acetylation events that are occurring at low stoichiometry, which most likely are not of real physiological significance—at least if the sites have a loss-of-function effect.

Reviewer #2 (Remarks to the Author):

This work by Kienle et al. is a systematic study on the acetylation of ubiquitin. Ubiquitination is a common post-translational modification of proteins that alters their structural and functional properties. Recent work by several groups have now illustrated that ubiquitin itself can be further modified by phosphorylation and acetylation, but little is known about the consequences of acetylation on ubiquitin. Here, the authors use genetic code expansion to make 7 different acetylated ubiquitin proteins (one at each of the 7 lysines). In agreement with other studies, acetylated ubiquitin affects the degree of E3 autoubiquitination. Using NMR spectroscopy, they characterize the structural effects of acetylation and find that each of the acetylated ubiquitins perturbs chemical shifts of backbone amides throughout ubiquitin, similar to how ubiquitin attachment at each of these lysines alters chemical shifts. Besides the structural characterization, the authors further study the acetylated ubiquitin interactome by mass spectrometry with HEK293T cell extracts, and use orthogonal experiments (Strep-Ub pulldowns) to identify proteins such as NDP52 and UCHL3 that specifically recognize certain forms of acetylated ubiquitin. Lastly, they identify p300 as a putative modulator of ubiquitin acetylation, particularly at lysine positions 11 and 48. However, it is important to note that acetylation affects a small portion of the overall ubiquitin population. There is much that remains to study of ubiquitin acetylation, specifically the different biological pathways in which ubiquitin acetylation is involved. This study begins to scratch the surface of another level of ubiquitin regulation. Some interesting ideas are provoked here, specifically how proteasomes may recognize K48-acetylated ubiquitin, thus not requiring a K48-linked polyUb chain to target a substrate for degradation. I do have suggestions below to further improve this study, as well as comments to address completeness. Given the authors' extensive characterization of the acetylated Ub mutants, I think it would be worth to examine the interaction between NDP52 and acetylated Ub more closely on a molecular level. This work is generally well-written and suitable for publication after the comments below are addressed.

Major comments:

- 1) Concerning the work with NDP52, I had the following suggestions. This is an interesting protein with purported selectivity towards K6 acetylation of Ub. Could the authors include the part of the sequence that was removed for the delta-ZF construct? Additionally in Figure 6e, what is the band immediately underneath the monoUb band? It would be helpful to include the original western blots here with a molecular weight marker. Along these lines, the authors mention that the K6R mutant did bind well to NDP52, but they mention "(data not shown)" for this. In light of how NDP52 is highlighted as a selective binder to K6ac Ub, this is important for elaboration. Could the authors perform binding affinity

experiments on NDP52? Given the large size of the protein, it may not be possible to use NMR spectroscopy but possibly other biophysical methods like ITC as done in (Xie et al., Autophagy 2015). If the full-length protein is not possible, perhaps an assay involving the ZF2 domain of NDP52 and K6ac Ub/K6R Ub/mono Ub. It would be preferable to do this without GST tags.

2) The authors find that the proximal Ub CSPs of Ub2 from Castaneda et al. 2016 mirror many of the CSPs for acetylated Ub. In another work from the Fushman lab (Castaneda et al., Structure 2016 <https://doi.org/10.1016/j.str.2016.01.007>), a comparison was made between monoUb and Lys-Boc-modified Ub, where Boc is a removable lysine protecting group. Lys-Boc mimics much of an acetylated lysine, and the Lys-Boc vs. monoUb CSPs mirror the Ac-K vs. monoUb CSPs in this work. See Figure 2 of Castaneda et al., Structure 2016. Comments on this work should be included here, and can be used as additional support for Kienle et al.

3) It is striking that K11Ac Ub has unique CSPs. On pg. 8, the authors propose that the length of the 11 side chain plays a role in eliciting steric effects on the alpha-helix. The authors should include mention of the Lys-Boc work as described above, as it supports their hypothesis given that Lys-Boc is a longer side chain than AcK.

4) In Figure 4, it is interesting to see that the CSPs between 11AcK and UQ2-UBA are identical at the titration endpoint compared to other Ub mutants and UBA. Despite this, the binding affinity for 11AcK is weaker than the other Ub mutants. Given that the starting concentration for the 11AcK Ub titration was 33 uM while the other Ubs were 150 uM (as noted in methods), could the authors do another control UBA titration experiment with monoUb at 33 uM? This is requested also as the titration curves in Figure S5 appear stoichiometric (reach saturation at 1:1 Ub:UBA concentrations).

Other comments:

Figure 1: The 11A/11Q/11R labels on Figure 1e should be fixed.

Figure 7: It is difficult to see some of the experimental data points as the red error bars cover some of these points. On Figure 7B, it is also difficult to determine whether the decreases in acetylation for K11 are significant or not. As for experimental reproducibility, could more details about the 'independent' experiments be provided in methods?

The full western blots should be shown as a supplemental figure to complement data presented in Figure 6.

Reviewer #3 (Remarks to the Author):

Review of NCOMMS-21-43565-T

The manuscript by Kienle et al. reports the most in-depth study to date regarding the consequences of acetylation modifications to the ubiquitin (Ub) protein, which is itself a common post-translational protein modifier. A previous report by Ohtake et al. established that cellular human Ub is acetylated on multiple lysine residues (albeit only to a very small percentage of Ub) and that two (of 7 possible) forms of acetylated lysine Ub yield differential protein Ub-modification patterns compared to unmodified Ub in cell-free ubiquitination assays. However, prior to the current manuscript, (1) no Ub acetylation “writers” had been identified, and (2) the potential cellular consequences of Ub acetylation were largely unexplored. Further, how specific lysine acetylation might impact the structure of Ub, besides the obvious loss of charge, was unknown. Kienle et al. report significant progress on all three of these fronts, especially the effects of site-specific acetylation on the structure of Ub (Figures 2-4). Figures 5-7 of the manuscript address the key question—does Ub acetylation have a functional consequence to human biology? Though this question remains mostly unanswered, the authors made significant progress by identifying proteins from HEK293T cell extracts that interact differentially with specific acetylated forms of Ub (i.e., potential “readers” of these specific modifiers) and by showing that p300 is a likely Ub acetylation writer. Since p300 has so many substrates and functions, it will likely be difficult to pinpoint its potential specific functions related to Ub acetylation. Further exploration is also needed to understand the first edition of the ‘acetylated ubiquitin human protein interactome’ generated by this manuscript. Still, I think the current manuscript is noteworthy, very well written, and near ready for publication. I have just a few additional, more specific comments, mainly for addition to the discussion section and for potential future directions.

Specific comments

1) The authors write “An attractive alternative (to K to Q mutagenesis) is to apply the genetic expansion method in eukaryotic cells”. This idea, and other musings, make me curious as to whether Ub acetylation occurs in non-mammalian eukaryotes. Plants? Fungi? *S. cerevisiae* is arguably the best-studied eukaryote with regard to ubiquitin biology and may be a good host for genetic expansion. Also, assuming Ub acetylation occurs in other eukaryotes, a very straightforward follow up study (beyond the scope of this manuscript) could compare the ‘acetylated ubiquitin interactomes’ across species. A short discussion of potential conservation for Ub acetylation might broaden interest in this work/topic.

2) Related (and also beyond the scope of the current manuscript), it would be interesting to profile acetylated ubiquitin interaction partners using different human cell types and after different cell growth conditions. There is potentially much more to learn!

3) With the different purified acetylated ubiquitin proteins in hand, a future direction could be to develop antibodies that are specific to particular modified proteins. These tools might then aid in discovering potential physiological roles for acetylated ubiquitin: immunoblotting under various treatments/conditions and with different cell types; immunoprecipitation, with the potential for downstream proteomics; immunofluorescence for subcellular localization. Perhaps the low levels of acetylated ubiquitin will be problematic for antibody-based detection but immunoblotting after various cellular treatments may reveal cellular conditions under which a higher percentage of ubiquitin is modified. Also, proteasome and DUB inhibitors could be explored to potentially “catch” acetylated ubiquitin species that may be short-lived.

-Christopher Hickey, PhD

Point-by-point response to reviewers' comments on "Electrostatic and steric effects underlie acetylation-induced changes in ubiquitin structure and function" (NCOMMS-21-43565-T)

Reviewer #1

The lab of Martin Scheffner presents a study to elucidate the role of post-translational lysine acetylation for ubiquitin function. Based on quantitative mass-spectrometry-based screens it was discovered that except from K29 all lysines in ubiquitin can be lysine acetylated. Kienle and co-workers present a thorough analysis of all of the six lysine-acetylation sites in ubiquitin and how acetylation affects structure and function. They used an elegant system, namely, genetic code expansion, to prepare site-specifically lysine-acetylated ubiquitin variants. To this end, they analyzed Ub-chain generation by E1 and selected E2 and E3 enzymes. Moreover, they performed NMR studies to analyze the impact of lysine acetylation on Ubiquitin structure and conformational changes that might affect the interplay with other proteins. These studies revealed that acetylation resulted in alterations in the Ub structure mainly by electrostatic effects by neutralization of the positive charge at the lysine side chain. Moreover, steric effects upon acetylation also contribute to the observed alterations in the Ub structure. Moreover, Kienle and co-workers performed pull-down experiments coupled to MS-based experiments with the acetylated ubiquitin variants to identify proteins for which the interaction was modulated in an acetylation-dependent manner. These experiments revealed proteins that interacted dependent on the presence of an acetyl-group at the various lysine side chains, others that were decreased in the interaction. Some interactions were modified with several acetylations in ubiquitin, while others were modulated specifically dependent on the presence of a specific lysine acetylation site. Among these interactors the authors identified DUBs, components of the 19S regulatory particle of the 26S proteasome. Moreover, these analyses revealed that Ub 6AcK binds specifically to the protein NDP52, which is involved in xenophagy of invading bacterial pathogens. Pull-down experiments confirmed this interaction and showed that the interaction depends on the presence of the acetyl-group at K6 in Ub. Finally, the authors did first studies on the regulation of ubiquitin lysine acetylation by lysine acetyltransferases p300/CBP. The authors performed *in vitro* acetylation of ubiquitin with subsequent MS analysis to show that mono-ubiquitin and Met1-linked di-ubiquitin are acetylated at K48 and K11. Overall, the manuscript presents interesting data on the role of lysine-acetylation on ubiquitin structure and function. The experiments performed were all of high quality. However, from my perspective the physiological significance of the findings are not elucidated enough to justify publication in Nature Communication. In the following points I summarize my major points of criticism:

1. The authors used selected E3 ligases to analyze the impact of the lysine-acetylated ubiquitin variants on ubiquitin chain formation. To this end, they selected HDM2, RLIM, and E6AP/UBE3A and analyzed the autoubiquitylation activity using the acetylated variants as substrates. This selection of E3 enzymes seems to be a bit random and only shows that efficiencies for these E3 ligases is affected not/affected. The authors conclude from their results that acetylation at K6, K11, K27 and K48 interferes with efficient ubiquitin chain formation. From my perspective, it is not possible to draw general conclusions from these results, i.e. how the activities of E1, E2 and of the app.600 E3 Ub-ligases are affected by ubiquitin acetylation at a specific site.

We agree with the reviewer that the selection of E3 ligases may appear to be a bit random

and that the data obtained cannot necessarily be extrapolated to other E3 ligases. In fact, we deliberately refrained from drawing general conclusions about the "general" usage of acetylated Ub variants (e.g. in the respective Results section, we are referring to the "E3s used" and "E3s tested"). Moreover, similar studies were published by other groups employing different E3 ligases as well as studying the usage of acetylated Ub variants by the E1 enzyme and a few E2 enzymes (refs. 18, 19, 47 of the revised manuscript). Thus, the main purpose of determining the usage of acetylated Ub variants by the E3s used was to corroborate the notion that the different variants have different structural and functional properties rather than making a general statement about the usage of the Ub variants by enzymes of the Ub conjugation system. Having said this, we hope that it is easy to understand why we used mainly HDM2 and E6AP for our studies. Firstly, our group has a long-standing interest into HDM2 and E6AP, which belong to the best-studied E3s in general, since their deregulation is causally associated with human disease. Secondly, HDM2 belongs to the RING family of E3s, while E6AP is a member of the HECT E3s. In other words, the E3s used by us, while "random", are "prototypal" and as representative as any other E3 that we could have used instead.

Maybe the authors should focus on specific cellular processes and elucidate the role of individual acetylation events in ubiquitin, including the analysis of the acetylation stoichiometry and the regulation by acetyltransferases (or non-enzymatic acetylation) and by deacetylases.

We fully agree that it will be important to study the role of acetylated Ub variants in cells. In fact, we are convinced that our interactome analysis is a crucial first step in this direction. With respect to stoichiometry, Ohtake et al. (ref. 19 of the manuscript) reported that in a particular cell line (derived from HEK293 cells that we used in our study), the percentage of acetylated Ub variants is rather low (0.01-0.03 %) but comparable to M1-linked Ub chains and that the percentage increases in the presence of general inhibitors of deacetylases indicating that Ub acetylation is subject to regulation. We hope that the reviewer agrees that our profound structural and interactome analyses of the various acetylated Ub variants pave the way for detailed cellular analyses, but that such studies are beyond the scope of the present manuscript. See also our response to comment 7 below.

2. The experiments analyzing the impact of K to Q, K to R and K to A mutants on ubiquitin chain formation by different E3 ligases are only done once and no quantification of the immunoblotting/Coomassie brilliant blue stained SDS-PAGE gels is performed, which make it really challenging to judge and compare the efficiencies of the reactions.

All ubiquitylation assays were performed at least three times with similar results. We apologize that this information was not provided in the first place (a respective sentence has been added to the Methods section).

Autoubiquitylation reactions were generally monitored by Coomassie blue staining. The relative efficiency of the reactions can be most easily compared by looking at the levels of free Ub (as indicated on page 6, 2nd paragraph). We deliberately did not quantify the respective levels, as the assay per se is not quantitative (i.e. relative differences are scored). However, if the reviewer feels this to be important, we will be happy to do so.

3. The authors show that for some mutants the K to Q mutant, often used as a mimic of lysine-acetylation cannot mimic lysine acetylation. Moreover, also the K to R mutants, often

used to conserve the positive charge to mimic an non-acetylated lysine does influence ubiquitin chain formation. These data show that electrostatic as well as steric effects are exerted by lysine acetylation to regulate ubiquitin function. As an example, the authors state, that for K6, the positive charge is the major driver for observed repressive effect on E3 ligase function. However if looking at the SDS-PAGE gels (Supp. Fig. 2) it is shown that Ub K6R can be used as substrate for E6AP but compared with Ub WT, the efficiency is clearly reduced (after 30 min there is still some mono-ubiquitin visible). This shows that next to charge neutralization also steric effects are essential to recover the full impact of lysine acetylation.

We are not sure what the reviewer is referring to. On page 5/6, we state "similar results were obtained with Ub variants modified at position 6, 11 or 27 for both E3s, insofar as a positive charge at these positions appears to be an important determinant for efficient autoubiquitylation (Fig. 1e and Supplementary Fig. 2)" and go on by stating "the above results clearly indicate that acetylation of K6, K11, K27, and K48 interferes with efficient Ub chain formation, which is, at least in part, caused by charge neutralization". In our opinion, these statements do not exclude that steric effects contribute to the effects observed. We rather focused on a more detailed discussion of K11, since for Ub 11AcK the steric effect can be most readily explained. To make this more clear, we have rephrased the second paragraph on page 6 (*For K11, this effect is explained on a structural level by the fact that the side chain amino group is involved in a salt bridge with E34. However, features other than charge neutralization appear to be involved as well. For instance, Ub 11A and Ub 11Q are more efficiently used by HDM2 for autoubiquitylation than Ub 11AcK (free Ub 11A and Ub 11Q are completely consumed after a reaction time of 150 min, while free Ub 11AcK can still be observed; Fig. 1e). Moreover, the results obtained with the Ub 11 variants also suggest that the frequently used AcK surrogate Q does not reliably mimic all inherent features of AcK.*)

4. The fact that K to Q and K to R mutants are no perfect tools to study lysine acetylation is not new and there are several publications showing that these can be poor tools to study lysine acetylation.

We fully agree with the reviewer that we are not the first to indicate that K-Q / K-R mutants are not always suited to study lysine acetylation. Admittedly, we are no "acetylation experts", so it is well possible that it has escaped our attention, but we are not aware of any publication, where respective mutants were compared side-by-side with truly site-specific acetylated proteins. Thus, if the reviewer provides us with respective citations, this will be much appreciated.

5. The finding that Ub 6AcK increases the binding towards NDP52 is really interesting as this means a "gain-of-function" for post-translational lysine acetylation. This is rarely observed for this modification. However, the interaction is analyzed only qualitatively and more quantitative data for the interaction would be desirable.

Thank you for this helpful suggestion. To obtain more quantitative data, we resorted to an ELISA-like approach (NMR and other biophysical methods such as bilayer interferometry did not work out for full-length NDP52). This revealed that Ub 6AcK binds approximately 8-fold better to NDP52 than non-modified Ub does (Supplementary Fig. 6). Due to precipitation

issues (NDP52 and/or the dye used for detection), saturation of the binding reaction could not be reached and, thus, actual K_d values could unfortunately not be determined.

6. Data on the regulation of the different Ub acetylation sites are almost completely missing. This is one of the major points of criticism. To show how the sites are acetylated (enzymatically and/or non-enzymatically) and deacetylated (enzymatically deacetylated or not) would be an important criterion to judge the physiological significance of these modifications. The only data the authors show is on interactome of ubiquitin, where HDAC6 is found, and on CBP/p300. P300/CBP is only capable to acetylate K48 and K11 in ubiquitin. How the other lysine side chains are acetylated is not analyzed further. The authors did not analyze any of the other known acetyltransferases and deacetylases.

We agree with the reviewer that it will be important to delineate the enzymes involved in regulating the acetylation status of Ub in a position-specific manner in cells. To start to address this issue, we show in the revised manuscript that HDAC6 does not only bind all acetylated Ub variants, but that is also capable of deacetylating all of these to a similar extent (Fig. 7). This suggests that unlike p300/CBP, HDAC6 plays a more general role in determining the acetylation status of Ub. Notably, we did not observe any spontaneous deacetylation of the Ub variants under the conditions used. Such studies can of course be extended to any acetyltransferase/deacetylase. However, we feel that this is clearly beyond the scope of this manuscript.

7. A major point of criticism is the physiological importance of the observed results. If the Ub acetylation has a loss-of-function effect on E3 mediated Ub chain formation etc. it is necessary that the acetylation reached sufficiently high stoichiometries. The authors should perform studies to show under which physiological conditions the acetylation of ubiquitin is increased and which stoichiometries are reached. This is essential to judge the physiological significance of a specific lysine acetylation event. Along that line, the authors do not present a single immunoblotting using an anti-acetyl lysine antibody that shows the acetylation of ubiquitin. Mass-spectrometry based methods are highly sensitive and will also show acetylation events that are occurring at low stoichiometry, which most likely are not of real physiological significance—at least if the sites have a loss-of-function effect.

As indicated in the Introduction of the manuscript (refs. 17-19), published data indicate that the acetylation status of at least some lysine residues of Ub is altered in cells in response to certain stimuli, but we fully agree that additional studies are clearly required to substantiate such data. Pan-specific anti-acetyl lysine antibodies do not appear to be suited for such studies, as they do not allow to distinguish acetylated Ub from other acetylated proteins. This is why we did not perform such experiments. Thus, an important task in the future will be the generation of antibodies that specifically recognize acetylated Ub in a position-specific manner. The possibility of generating large amounts of recombinant Ub AcK variants should prove highly beneficial for such endeavors.

As discussed in our response to comment 1, levels of acetylated Ub variants in cells are rather low. This, however, does not exclude the possibility that for instance under certain conditions, levels of a distinct acetylated Ub variant are highly increased at a distinct location in a cell and, thus, loss-of-function effects can be observed. Nonetheless, we believe that gain-of-function effects are more likely and/or can be more easily scored for. This notion is supported by our interaction analyses, where we identified proteins that bind more efficiently to acetylated Ub variants than to non-modified Ub rather than the other way around.

Reviewer #2:

This work by Kienle et al. is a systematic study on the acetylation of ubiquitin. Ubiquitination is a common post-translational modification of proteins that alters their structural and functional properties. Recent work by several groups have now illustrated that ubiquitin itself can be further modified by phosphorylation and acetylation, but little is known about the consequences of acetylation on ubiquitin. Here, the authors use genetic code expansion to make 7 different acetylated ubiquitin proteins (one at each of the 7 lysines). In agreement with other studies, acetylated ubiquitin affects the degree of E3 autoubiquitination. Using NMR spectroscopy, they characterize the structural effects of acetylation and find that each of the acetylated ubiquitins perturbs chemical shifts of backbone amides throughout ubiquitin, similar to how ubiquitin attachment at each of these lysines alters chemical shifts. Besides the structural characterization, the authors further study the acetylated ubiquitin interactome by mass spectrometry with HEK293T cell extracts, and use orthogonal experiments (Strep-Ub pulldowns) to identify proteins such as NDP52 and UCHL3 that specifically recognize certain forms of acetylated ubiquitin. Lastly, they identify p300 as a putative modulator of ubiquitin acetylation, particularly at lysine positions 11 and 48. However, it is important to note that acetylation affects a small portion of the overall ubiquitin population. There is much that remains to study of ubiquitin acetylation, specifically the different biological pathways in which ubiquitin acetylation is involved. This study begins to scratch the surface of another level of ubiquitin regulation. Some interesting ideas are provoked here, specifically how proteasomes may recognize K48-acetylated ubiquitin, thus not requiring a K48-linked polyUb chain to target a substrate for degradation. I do have suggestions below to further improve this study, as well as comments to address completeness. Given the authors' extensive characterization of the acetylated Ub mutants, I think it would be worth to examine the interaction between NDP52 and acetylated Ub more closely on a molecular level. This work is generally well-written and suitable for publication after the comments below are addressed.

Major comments:

1) Concerning the work with NDP52, I had the following suggestions. This is an interesting protein with purported selectivity towards K6 acetylation of Ub. Could the authors include the part of the sequence that was removed for the delta-ZF construct? Additionally in Figure 6e, what is the band immediately underneath the monoUb band? It would be helpful to include the original western blots here with a molecular weight marker. Along these lines, the authors mention that the K6R mutant did bind well to NDP52, but they mention "(data not shown)" for this. In light of how NDP52 is highlighted as a selective binder to K6ac Ub, this is important for elaboration. Could the authors perform binding affinity experiments on NDP52? Given the large size of the protein, it may not be possible to use NMR spectroscopy but possibly other biophysical methods like ITC as done in (Xie et al., Autophagy 2015). If the full-length protein is not possible, perhaps an assay involving the ZF2 domain of NDP52 and K6ac Ub/K6R Ub/mono Ub. It would be preferable to do this without GST tags.

We apologize that we did not make this more clear in the first place. In the delta-ZF construct, the C-terminal 32 amino acids of NDP52 are deleted. This information is now provided in Fig. 6d and in the Methods section. We do not know what the band underneath the Ub band is in Fig. 6e. However, as it is present in all samples, it most likely represents an unspecific artefact that is not relevant (Fig. 1 for reviewer 2).

Fig. 1 for reviewer 2. Complete western blot of Fig. 6e.

To obtain more quantitative data on the NDP52 interaction, we resorted to an ELISA-like approach (NMR and other biophysical methods such as bilayer interferometry did not work out). This revealed that Ub 6AcK binds approximately 8-fold better than non-modified Ub and 2-fold better than Ub 6R to NDP52 (Supplementary Fig. 6). Due to precipitation issues at higher concentrations (NDP52 and/or the dye used for detection), saturation of the binding reaction could not be reached and, thus, actual K_d values could not be determined. Similar, we had to use GST fusion proteins of NDP52 for the analysis (we also tested a Strep-tagged version of NDP52; however, its detection by an anti-Strep tag antibody was not sensitive enough for the ELISA-like approach), but since we used the delta-ZF2 mutant as control, we hope that the reviewer agrees that this is of no concern. The data have been added to the manuscript (page 13; Supplementary Fig. 6j)

2) The authors find that the proximal Ub CSPs of Ub2 from Castaneda et al. 2016 mirror many of the CSPs for acetylated Ub. In another work from the Fushman lab (Castaneda et al., Structure 2016 <https://doi.org/10.1016/j.str.2016.01.007>), a comparison was made between monoUb and Lys-Boc-modified Ub, where Boc is a removable lysine protecting group. Lys-Boc mimics much of an acetylated lysine, and the Lys-Boc vs. monoUb CSPs mirror the Ac-K vs. monoUb CSPs in this work. See Figure 2 of Castaneda et al., Structure 2016. Comments on this work should be included here, and can be used as additional support for Kienle et al.

and

3) It is striking that K11Ac Ub has unique CSPs. On pg. 8, the authors propose that the length of the 11 side chain plays a role in eliciting steric effects on the alpha-helix. The authors should include mention of the Lys-Boc work as described above, as it supports their hypothesis given that Lys-Boc is a longer side chain than AcK.

Thank you for pointing this out. The work by Castaneda et al. is now referenced (ref. 39) and briefly (because of length constraints) discussed in the text (pages 9 and 16, first paragraph)

4) In Figure 4, it is interesting to see that the CSPs between 11AcK and UQ2-UBA are identical at the titration endpoint compared to other Ub mutants and UBA. Despite this, the binding affinity for 11AcK is weaker than the other Ub mutants. Given that the starting concentration for the 11AcK Ub titration was 33 μ M while the other Ubs were 150 μ M (as noted in methods), could the authors do another control UBA titration experiment with monoUb at 33 μ M? This is requested also as the titration curves in Figure S5 appear stoichiometric (reach saturation at 1:1 Ub:UBA concentrations).

Thank you for bringing this to our attention. As the curvature of binding isotherms is intimately depending on the protein concentration, we generally share the reviewer's concerns that this might be a potential source for overestimating the K_d value. Thus, we repeated the titration experiment with non-modified Ub by applying the same experimental conditions as used for Ub 11AcK with respect to concentration at the start of the titration experiment ($c = 33 \mu\text{M}$) and molar ratios between UQ2-UBA and non-modified Ub. Under these conditions, the K_d value of non-modified Ub was determined to be $2.0 \pm 0.1 \mu\text{M}$. Although this value is higher than the one obtained with a starting concentration of Ub of $157 \mu\text{M}$ ($K_d = 0.5 \pm 0.6 \mu\text{M}$), the binding affinity of UQ2-UBA to Ub 11AcK is still significantly reduced ($K_d = 8.0 \pm 0.8 \mu\text{M}$). Both CSP mapping and binding isotherms of the replicated experiment using non-modified Ub were added to Fig. 4, while the corresponding K_d values for the respective Ub variants are provided in Supplementary Fig. 5.

Other comments:

Figure 1: The 11A/11Q/11R labels on Figure 1e should be fixed.

Thank you for spotting this error, which has been fixed.

Figure 7: It is difficult to see some of the experimental data points as the red error bars cover some of these points. On Figure 7B, it is also difficult to determine whether the decreases in acetylation for K11 are significant or not. As for experimental reproducibility, could more details about the 'independent' experiments be provided in methods?

We modified the presentation and hope that the data are clearer now. For K11, acetylation was observed only with linear Ub dimers and only in the presence of both p300 and Acetyl-CoA. Acetylation was studied in three independent experiments. A respective sentence was added to the Methods section.

For clarification, we would also like to mention that the K11-acetylated peptide was mainly detected in a triply protonated state (indicated with "+++"), whereas the K48-acetylated peptide was mainly found in a doubly protonated state. These charge states are therefore depicted individually (for reasons of reproducibility for other groups), though they reflect the same acetylated peptide.

The full western blots should be shown as a supplemental figure to complement data presented in Figure 6.

The full Western blots are provided in the Source Data file.

Reviewer #3:

The manuscript by Kienle et al. reports the most in-depth study to date regarding the consequences of acetylation modifications to the ubiquitin (Ub) protein, which is itself a common post-translational protein modifier. A previous report by Ohtake et al. established that cellular human Ub is acetylated on multiple lysine residues (albeit only to a very small percentage of Ub) and that two (of 7 possible) forms of acetylated lysine Ub yield differential

protein Ub-modification patterns compared to unmodified Ub in cell-free ubiquitination assays. However, prior to the current manuscript, (1) no Ub acetylation “writers” had been identified, and (2) the potential cellular consequences of Ub acetylation were largely unexplored. Further, how specific lysine acetylation might impact the structure of Ub, besides the obvious loss of charge, was unknown. Kienle et al. report significant progress on all three of these fronts, especially the effects of site-specific acetylation on the structure of Ub (Figures 2-4). Figures 5-7 of the manuscript address the key question—does Ub acetylation have a functional consequence to human biology? Though this question remains mostly unanswered, the authors made significant progress by identifying proteins from HEK293T cell extracts that interact differentially with specific acetylated forms of Ub (i.e., potential “readers” of these specific modifiers) and by showing that p300 is a likely Ub acetylation writer. Since p300 has so many substrates and functions, it will likely be difficult to pinpoint its potential specific functions related to Ub acetylation. Further exploration is also needed to understand the first edition of the ‘acetylated ubiquitin human protein interactome’ generated by this manuscript. Still, I think the current manuscript is noteworthy, very well written, and near ready for publication. I have just a few additional, more specific comments, mainly for addition to the discussion section and for potential future directions.

Specific comments

*1) The authors write “An attractive alternative (to K to Q mutagenesis) is to apply the genetic expansion method in eukaryotic cells”. This idea, and other musings, make me curious as to whether Ub acetylation occurs in non-mammalian eukaryotes. Plants? Fungi? *S. cerevisiae* is arguably the best-studied eukaryote with regard to ubiquitin biology and may be a good host for genetic expansion. Also, assuming Ub acetylation occurs in other eukaryotes, a very straightforward follow up study (beyond the scope of this manuscript) could compare the ‘acetylated ubiquitin interactomes’ across species. A short discussion of potential conservation for Ub acetylation might broaden interest in this work/topic.*

We fully agree with all comments and suggestions of the reviewer. To our knowledge, proteomics studies show that Ub is acetylated in *S. cerevisiae*, while data for other organisms do not appear to be available. We added a sentence about yeast to the Discussion, page 17 last paragraph).

2) Related (and also beyond the scope of the current manuscript), it would be interesting to profile acetylated ubiquitin interaction partners using different human cell types and after different cell growth conditions. There is potentially much more to learn!

and

3) With the different purified acetylated ubiquitin proteins in hand, a future direction could be to develop antibodies that are specific to particular modified proteins. These tools might then aid in discovering potential physiological roles for acetylated ubiquitin: immunoblotting under various treatments/conditions and with different cell types; immunoprecipitation, with the potential for downstream proteomics; immunofluorescence for subcellular localization. Perhaps the low levels of acetylated ubiquitin will be problematic for antibody-based detection but immunoblotting after various cellular treatments may reveal cellular conditions under which a higher percentage of ubiquitin is modified. Also, proteasome and DUB

inhibitors could be explored to potentially “catch” acetylated ubiquitin species that may be short-lived.

We very much appreciate these suggestions that provide us with a great road map for further studies. For instance, as indicated in the Discussion, we currently explore the possibility to employ non-hydrolyzable acetyllysine analogs to study the effect of protein acetylation *in vitro* and in cells. In addition, Ub variants containing such non-hydrolyzable analogs should also prove helpful in the generation of antibodies that specifically recognize acetylated Ub in a position-specific manner. We would be happy to extend the Discussion of our results along the reviewer's suggestions; however, due to space constraints we feel that this cannot be done with the depth required.

REVIEWER COMMENTS

Reviewer #1 (Remarks to the Author):

Review to the revised manuscript "Electrostatic and steric effects underlie acetylation-induced changes in ubiquitin structure and function(NCOMMS-21-43565-T)"

The manuscript improved, however, there are still some critical points, which are clarified here:

1. Stoichiometry of acetylation: The authors state that determination of the acetylation stoichiometry is beyond the scope of the current manuscript. However, the stoichiometry is the important factor for the physiological outcome of the acetylation of ubiquitin. Apart from the interaction with NDP52, which seems to be improved upon acetylation of ubiquitin, all other effects observed on Ub-chain formation are loss-of-function effects. For these effects high stoichiometries of acetylation are needed to observe the effect. To say drastically, the cell might not care if there is less than 1% acetylated at a lysine side chain on a Ub, if more than 99% are not acetylated. If using the genetic code expansion you work with quantitatively and homogeneously acetylated, i.e. 100% acetylated and only at one individual site acetylated protein. The situation in vivo might be different, i.e. the stoichiometry is likely less than 100% and an Ub might be acetylated at various sites. The stoichiometry might be high under specific cellular conditions, i.e. high acetyl-CoA levels (if acetylation is also non-enzymatically driven), expression levels and activities of KATs/KDACs/sirtuins (availability of NAD⁺). Moreover, if on a Ub-chain there is acetylation at a lower stoichiometry on each single Ub molecule, it might add up to create a significant impact. From my point of view these are essential points to at least discuss.

2. Reliability of Q/R-mutants: there are several papers out that show that Q/R-mutants are sometimes no reliable tools to study lysine acetylation. It very much depends on the individual protein and on the individual site of acetylation if this can be studied reliably by K to Q and K to R. To this end, genetically encoding acetyl-lysine is such a powerful system to study the real consequences of lysine acetylation. See as an examples:

Knyphausen P, Lang F, Baldus L, Extra A, Lammers M. 2016. Insights into K-Ras 4B regulation by post-translational lysine acetylation. *Biol Chem* 397: 1071-85

de Boor S, Knyphausen P, Kuhlmann N, Wroblowski S, Brenig J, Scislowski L, Baldus L, Nolte H, Kruger M, Lammers M. 2015. Small GTP-binding protein Ran is regulated by posttranslational lysine acetylation. *Proc Natl Acad Sci U S A* 112: E3679-88

Okada AK, Teranishi K, Ambroso MR, Isas JM, Vazquez-Sarandeses E, Lee JY, Melo AA, Pandey P, Merken D, Berndt L, Lammers M, Daumke O, Chang K, Haworth IS, Langen R. 2021. Lysine acetylation regulates the interaction between proteins and membranes. *Nat Commun* 12: 6466

3. Using pan-anti-acetyl-lysine antibody: The authors are right in saying that it is not possible to narrow down the individual acetylation site using a pan-antibody. However, it would be good to see if it is at all possible to isolate acetylated ubiquitin from cells, i.e. by an immunoprecipitation. This would at least be an argument for acetylation being present at a detectable level by Western blot (by the way: Western blot is written with capital letter as it is derived from Southern blot and Southern was a name of a scientist). Treatment of cells with KDAC and/or sirtuin inhibitors would also at least qualitatively show if the acetylation level is regulated by KDACs/sirtuins. To find out which sites are affected a subsequent determination by mass spectrometry could be performed.

Reviewer #2 (Remarks to the Author):

The revised manuscript by Kienle et al. is improved with additional experiments to investigate general deacetylation of ubiquitin by HDAC. The authors also performed an ELISA assay to determine relative binding affinities between Ub/Ub-K6Ac and NDP52. The discussion is further modified to include additional references and speculations towards physiological/biological significance of Ub acetylation. The revised study does address this reviewer's concerns, particularly with binding affinity measurements involving NDP52. The authors correctly note that there is significant work still to be done to understand Ub acetylation, including the development of specific antibodies to recognize the PTM. I thank the authors for including more details in the methods section as well as including information on the reproducibility of the experiments. I support publication of the manuscript in its current form.

Point-by-point response to reviewers' comments on "Electrostatic and steric effects underlie acetylation-induced changes in ubiquitin structure and function" (NCOMMS-21-43565-T)

Reviewer #1

The manuscript improved, however, there are still some critical points, which are clarified here:

1. Stoichiometry of acetylation: The authors state that determination of the acetylation stoichiometry is beyond the scope of the current manuscript. However, the stoichiometry is the important factor for the physiological outcome of the acetylation of ubiquitin. Apart from the interaction with NDP52, which seems to be improved upon acetylation of ubiquitin, all other effects observed on Ub-chain formation are loss-of-function effects. For these effects high stoichiometries of acetylation are needed to observe the effect. To say drastically, the cell might not care if there is less than 1% acetylated at a lysine side chain on a Ub, if more than 99% are not acetylated. If using the genetic code expansion you work with quantitatively and homogeneously acetylated, i.e. 100% acetylated and only at one individual site acetylated protein. The situation in vivo might be different, i.e. the stoichiometry is likely less than 100% and an Ub might be acetylated at various sites. The stoichiometry might be high under specific cellular conditions, i.e. high acetyl-CoA levels (if acetylation is also non-enzymatically driven), expression levels and activities of KATs/KDACs/sirtuins (availability of NAD⁺). Moreover, if on a Ub-chain there is acetylation at a lower stoichiometry on each single Ub molecule, it might add up to create a significant impact. From my point of view these are essential points to at least discuss.

When considering the effect of acetylated Ub variants on Ub chain formation, we agree with the notion "*the stoichiometry is the important factor for the physiological outcome*". However, as indicated in our response to the comments of the reviewer on the original manuscript, we would like to emphasize that the main purpose of determining the usage of acetylated Ub variants by the Ub-conjugating system was to corroborate the notion that the different variants have different structural and functional properties, rather than implying that this (interference with Ub chain formation) is an important physiological function of Ub acetylation. Yet, as also indicated in our response to the comments on the original manuscript, this does not exclude the possibility that under certain conditions, levels of a distinct acetylated Ub variant are highly increased at distinct locations in a cell and, thus, physiologically relevant loss-of-function effects may be observed.

Nonetheless, we believe that gain-of-function effects are more likely or can at least be more easily scored for. This notion is supported by our interaction analyses, where we identified proteins that bind more efficiently to acetylated Ub variants than to non-modified Ub. In such a scenario, stoichiometry is less or even of no concern. For instance, ubiquitination comes in a multitude of flavors, including a virtually infinite number of differently linked Ub chains. One of these chains is M1-linked, the steady state level of which is in the range of 0.02 percent of total Ub. Importantly, Ohtake et al. (see also response to comment 3) have shown that e.g. K6-acetylated Ub accounts for approximately 0.03 percent of total Ub. The physiological relevance of M1-linked Ub chains is indisputable. Thus, considering gain-of-function effects it seems likely to us that site-specific acetylation of Ub has a significant impact not only on Ub structure (as we show) but also on Ub function. As abovementioned, our interaction analyses are in strong support of the latter. Moreover, acetylated Ub variants can exert their gain-of-functions effects

as free (i.e. unconjugated) molecules or as part of Ub chains. To make the issues of "stoichiometry" and "loss-of-function vs. gain-of-function" clearer, we have extended the respective paragraph in the Discussion (page 16, 1st paragraph).

2. Reliability of Q/R-mutants: there are several papers out that show that Q/R-mutants are sometimes no reliable tools to study lysine acetylation. It very much depends on the individual protein and on the individual site of acetylation if this can be studied reliable by K to Q and K to R. To this end, genetically encoding acetyl-lysine is such a powerful system to study the real consequences of lysine acetylation. See as an examples:

Knyphausen P, Lang F, Baldus L, Extra A, Lammers M. 2016. Insights into K-Ras 4B regulation by post-translational lysine acetylation. Biol Chem 397: 1071-85

de Boor S, Knyphausen P, Kuhlmann N, Wroblowski S, Brenig J, Scislowski L, Baldus L, Nolte H, Kruger M, Lammers M. 2015. Small GTP-binding protein Ran is regulated by posttranslational lysine acetylation. Proc Natl Acad Sci U S A 112: E3679-88

Okada AK, Teranishi K, Ambroso MR, Isas JM, Vazquez-Sarandeses E, Lee JY, Melo AA, Pandey P, Merken D, Berndt L, Lammers M, Daumke O, Chang K, Haworth IS, Langen R. 2021. Lysine acetylation regulates the interaction between proteins and membranes. Nat Commun 12: 6466

We are grateful for these helpful suggestions and apologize for our ignorance in this respect. We have changed the text including references at two occasions accordingly (page 6, 1st paragraph; page 17, 2nd paragraph).

3. Using pan-anti-acetyl-lysine antibody: The authors are right in saying that it is not possible to narrow down the individual acetylation site using a pan-antibody. However, it would be good to see if it is at all possible to isolate acetylated ubiquitin from cells, i.e. by an immunoprecipitation. This would at least be an argument for acetylation being present at a detectable level by Western blot (by the way: Western blot is written with capital letter as it is derived from Southern blot and Southern was a name of a scientist). Treatment of cells with KDAC and/or sirtuin inhibitors would also at least qualitatively show if the acetylation level is regulated by KDACs/sirtuins. To find out which sites are affected a subsequent determination by mass spectrometry could be performed.

Ohtake et al. (ref. 19 of the manuscript) performed IPs with an anti-Ub antibody (FK2) that is commonly used in the field and precipitates Ub conjugates in general and to a lesser extent also free "mono"-Ub. They subjected the material to SDS-PAGE, cut the respective lanes into three slices (proteins with a molecular mass of >90 kDa, 15-90 kDa, and <15 kDa), trypsinized the gel slices, and analyzed the eluates by quantitative high-resolution mass spectrometry. By this, they determined the levels of Ub acetylated at K6 (0.03 percent) and K48 (0.01 percent) in comparison to total Ub levels. Moreover, by using inhibitors of different classes of deacetylases, the authors provided good evidence that Ub acetylation is a dynamic process and that deacetylation of Ub is a redundant process insofar as different classes of deacetylases appear to be involved (page 3, last paragraph of our manuscript). The data are of high quality and reliable and together with proteomic studies (e.g. indicating that p300 is involved in acetylation of Ub at K11 and K48; ref. 21 of our manuscript and mentioned at several occasions; showing that rapamycin-induced autophagy affects acetylation of Ub at K6, page 12, last paragraph, ref. 48 of our manuscript) clearly indicate that Ub acetylation is subject to regulation in cells. We

would also like to emphasize that the cell line used by Ohtake et al. as well as by us (HEK293 cells) contains on average approximately 8×10^7 Ub molecules per cell (Kaiser SE et al., 2011, Nature Methods 8, pp. 691); i.e. 0.03 percent Ack6-Ub equals 24.000 molecules, supporting the "gain-of-function" possibility. We deliberately refrained from a detailed discussion of the study of Ohtake et al., as it would not add to the main messages of our study, but rather focused on a brief discussion of "loss-of-function" vs. "gain-of-function" effects (see above, response to comment 1). However, if the reviewer feels this to be important, we will be happy to do so.

We respectfully disagree with the statement "*This would at least be an argument for acetylation being present at a detectable level by Western blot*", mainly for two reasons. Firstly, the sensitivity of a Western blot analysis depends exclusively on the sensitivity of the antibody available. In other words, if (a modified form of) a protein is not or only poorly detectable in a Western blot analysis, this does not mean that it is not present in "physiologically relevant" levels; it may just indicate that the sensitivity of the antibody is not good enough. Secondly, and this relates to the sensitivity issue, we would have to analyze free "mono"-Ub to clearly show that Ub itself is indeed acetylated and not a protein modified by Ub. Ub is one of the evolutionally most conserved proteins. Thus, it is rather difficult, if not impossible to raise antibodies that recognize free mono-Ub with a sensitivity that would be required to detect low abundant forms of Ub in a reliable manner. Furthermore, quantification of Ub levels by Western blot analysis is not possible (due to the many different forms of Ub), which is one of the reasons for why the field is using sophisticated mass spectrometric approaches to do so (e.g. Kaiser SE et al., 2011, Nature Methods 8, pp. 691). Having said this and in view of the data by Ohtake et al. and the available proteomics data, we hope that the reviewer agrees that repeating such elaborate and time-consuming procedures would most certainly not result in any new data and, more importantly, would not improve the quality of our manuscript.

We thank the reviewer for indicating the typing error (Western blot). We have corrected it at all occasions.

Reviewer #2:

The revised manuscript by Kienle et al. is improved with additional experiments to investigate general deacetylation of ubiquitin by HDAC. The authors also performed an ELISA assay to determine relative binding affinities between Ub/Ub-K6Ac and NDP52. The discussion is further modified to include additional references and speculations towards physiological/biological significance of Ub acetylation. The revised study does address this reviewer's concerns, particularly with binding affinity measurements involving NDP52. The authors correctly note that there is significant work still to be done to understand Ub acetylation, including the development of specific antibodies to recognize the PTM. I thank the authors for including more details in the methods section as well as including information on the reproducibility of the experiments. I support publication of the manuscript in its current form.

We were glad to learn that that the reviewer supports publication of our manuscript and thank the reviewer again for helping us to improve the quality of our manuscript.